# The effect of overshooting 1.5°C global warming on the mass loss of the Greenland Ice Sheet

Martin Rückamp[1], Ulrike Falk[2], Katja Frieler[3], Stefan Lange[3], and Angelika Humbert[1,4]

[1]Alfred Wegener Institute, Helmholtz Centre for Polar and Marine Research, Bremerhaven, Germany
[2]formerly Alfred Wegener Institute, Helmholtz Centre for Polar and Marine Research, Bremerhaven, Germany
[3]Potsdam Institute for Climate Impact Research, Potsdam, Germany
[4]University of Bremen, Bremen, Germany

*Correspondence to:* martin.rueckamp@awi.de

**Abstract.** Sea-level rise associated with changing climate is expected to pose a major challenge for societies. Based on the efforts of COP21 to limit global warming to 2.0°C or even 1.5°C by the end of the 21[th] century (Paris Agreement), we simulate the future contribution of the Greenland ice sheet (GrIS) to sea-level change under the low emission representative concentration pathway (RCP) 2.6 scenario. The ice sheet system model (ISSM) with higher order approximation is used and
5    initialized with a hybrid approach of spin-up and data assimilation. For three general circulation models (GCMs: HadGEM2-ES, IPSL-CM5A-LR, MIROC5) the projections are conducted up to 2300 with forcing fields for surface mass balance (SMB) and ice surface temperature ($T_s$) computed by the surface energy balance model of intermediate complexity (SEMIC). The projected sea-level rise ranges between 21–38 mm by 2100 and 36–85 mm by 2300. According to the three GCMs used, global warming will exceed 1.5°C early in the 21[th] century. The RCP2.6 peak and decline scenario is therefore manually adjusted
10    in another set of experiments to suppress the 1.5°C-overshooting effect. These scenarios show a sea-level contribution that is on average about 38% and 31% less by 2100 and 2300, respectively. For some experiments, the rate of mass loss in the 23[rd] century does not exclude a stable ice sheet in the future. This is due to a spatially-integrated SMB that remains positive and reaches values similar to the present day in the latter half of the simulation period. Although the mean SMB is reduced in the warmer climate, a future steady-state ice sheet with lower surface elevation and hence volume might be possible. Our
15    results indicate that uncertainties in the projections stem from the underlying GCM climate data used to calculate the surface mass balance. However, the RCP2.6 scenario will lead to significant changes of the GrIS, including elevation changes of up to 100 m. The sea-level contribution estimated in this study may serve as a lower bound for the RCP2.6 scenario, as the currently observed sea-level rise is not reached in any of the experiments; this is attributed to processes (e.g. ocean forcing) not yet represented by the model, but proven to play a major role in GrIS mass loss.

# 1 Introduction

Within the past decade, the Greenland ice sheet has contributed by about 20% to sea-level rise (Rietbroek et al., 2016), caused by the acceleration of outlet glaciers and changes in the surface mass balance (Enderlin et al., 2014). In the past decades, these changes in surface mass balance contributed about 60% of the ice-sheet's mass loss, whereas 40% is attributed to increasing
discharge (van den Broeke et al., 2016). The question arises, which impact the GrIS will have on sea-level change in the next decades and centuries.

    Negotiated during COP21, the Paris Agreement's aim is to keep the global temperature rise in this century well below 2°C above pre-industrial levels and to pursue efforts to limit the temperature increase even further to 1.5 degrees Celsius (UNFCCC, 2015). However, the statement of holding global temperature below 2°C implies keeping global warming below the 2°C limit
over the course of the entire century and beyond, while efforts to limit the temperature increase to 1.5°C are often interpreted as allowing for a potential overshoot before returning to below 1.5°C (Rogelj et al., 2015). Here we selected the Representative Concentration Pathways (RCP, Moss et al., 2010) 2.6, it being the lowest emission scenario considered within CMIP5 and in line with a 1.5°C or 2°C limit of global warming. Depending on the general circulation models (GCM) considered, the global temperature change over time varies considerably, although the political target is met by 2100. Whereas some models
in RCP2.6 do not exceed the limit of 1.5°C or 2.0°C global warming before 2100, other models do and exhibit subsequent cooling (Frieler et al., 2017).

    While global temperature rise may be limited to 1.5°C or 2°C by 2100, warming over Greenland is enhanced due to the Arctic amplification (Pithan and Mauritsen, 2014), has already exceeded 1.5°C (relative to 1951–1980) in the past decade (GISTEMP Team, 2018) and may exceed 4°C by 2100. This yields more than 2°C warming by 2100 and could therefore have
a considerable impact on ice sheet mass loss over Greenland. This implies an enlargement of the ablation zone and goes along with a decline in SMB. However, it is currently unclear, how fast GrIS could react to cooling and recovery of SMB, as ice sheets are also reacting dynamically to atmospheric forcing.

    Recent large-scale ice sheet modeling attempts to project the contribution of the GrIS under RCP2.6 warming scenarios are very scarce. Fürst et al. (2015) conducted an extensive study to simulate future ice volume changes driven by both atmospheric
and oceanic temperature changes for all four representative concentration pathway scenarios. For the RCP2.6 scenario they estimate a sea-level contribution of 42.3±18.0 mm by 2100 and 88.2±44.8 mm by 2300. The value by 2100 is in line with estimates given by the Fifth Assessment Report of the Intergovernmental Panel on Climate Change (IPCC AR5, IPCC (2013)). The AR5 range for RCP2.6 lies between 10-100 mm by 2100 (the value depends on whether ice-dynamical feedbacks are considered or not).
The GrIS response to projections of future climate change are usually studied with a numerical ice sheet model (ISM) forced with climate data. ISM response is subject to the dynamical part and the surface mass balance (SMB). In the past, ISMs often used the rather simple and empirical based positive degree day (PDD) scheme, in which the PDD index is used to compute melt, run-off and ice surface temperature from atmospheric temperature and precipitation (Huybrechts et al., 1991). One disadvantage of the PDD method is, that the involved PDD parameters are tuned to correctly represent present-day melting

rates but may fail to represent past or future climates (Bougamont et al., 2007; Bauer and Ganopolski, 2017). On one far end of model complexity, a regional climate model (RCM) resolves most processes at the ice-atmosphere interface and in the upper firn layers, such as RACMO (Noël et al., 2018) or MAR (Fettweis et al., 2017) with higher spatial and temporal resolution than GCMs. RCMs have been shown to be quite successful in reproducing the current SMB of the GrIS. However, as they are computationally expensive, an intermediate way would be more efficient, balancing computational costs and parameterization of processes, such as the surface energy balance model of intermediate complexity (SEMIC, Krapp et al., 2017), which is employed in this study.

Here we target RCP2.6 peak and decline scenarios in particular to study the GrIS response to overshooting with a numerical ISM. The projections are driven with climate data output from the CMIP5 RCP2.6 scenario provided by the ISIMIP2b project for different GCMs (Frieler et al., 2017). To obtain ice surface temperature and surface mass balance from the atmospheric fields, the surface energy balance model SEMIC (Krapp et al., 2017) is applied. The SEMIC model (Sect. 2.1) is driven off-line to the ISM and therefore the climate forcing is one-way coupled and applied as anomalies to the ISM. The advantage of this one-way coupling is the lower computational cost, allowing for reasonably high spatial and temporal resolution of the ISM. In order to study the effect of overshooting, we design a RCP2.6-like scenario without overshoot by manually stabilizing the forcing at 1.5°C.

For modeling the flow dynamics and future evolution of the GrIS under RCP2.6 scenarios, the thermo-mechanical coupled Ice Sheet System Model (ISSM, Larour et al., 2012) with a Blatter-Pattyn type higher order momentum balance (BP; Blatter, 1995; Pattyn, 2003) is applied (Sect. 2.5). A crucial prerequisite for projections is a reasonable initial state of the ice sheet in terms of ice thickness, ice extent and ice velocities. Beside starting projections with the most realistic setting, the prevention of a model shock after switching from the initialization procedure to projections, is very important. Both have been a major issue in the past, which gave rise to an international benchmark experiment, initMIP Greenland (Goelzer et al., 2018), for finding optimal strategies to derive initial states for the ice velocity and temperature fields. Here, we apply a hybrid approach between a thermal paleo-spin up and data assimilation.

Before driving the projections, the SMB forcing is validated thoroughly against RACMO. Then we explore the response of the GrIS and its contribution to sea-level rise under the RCP2.6 scenario and a modified RCP2.6 scenario without overshoot.

## 2 Model description

### 2.1 Energy Balance Model

Thermo-mechanical ISMs require the annual mean surface temperatures and annual mean surface mass balance of ice as boundary conditions at the surface. To derive these ice sheet specific quantities, we use the surface energy balance model of intermediate complexity (SEMIC, Krapp et al., 2017). Although we only apply SEMIC and do not adjust parameters of SEMIC, SEMIC is described briefly. SEMIC computes the mass and energy balance of snow and/or ice surface. In order to

tune parameters for a number of processes, Krapp et al. (2017) performed an optimization for the GrIS forced with regional climate model data (MAR). These parameters have been used in our study, too. The energy balance equation reads as

$$c_{\mathrm{eff}} \frac{\mathrm{d}T_{\mathrm{s}}}{dt} = (1 - \alpha) \cdot \mathrm{SW}^{\downarrow} - \mathrm{LW}^{\uparrow} + \mathrm{LW}^{\downarrow} - H_{\mathrm{S}} - H_{\mathrm{L}} - Q_{\mathrm{M/R}} \,, \tag{1}$$

where $\alpha$ is the surface albedo that is parameterized with the snow height (Oerlemans and Knap, 1998). The downwelling shortwave $\mathrm{SW}^{\downarrow}$ and downwelling longwave radiation $\mathrm{LW}^{\downarrow}$ at the surface are provided as atmospheric forcing (sect. 2.2). The upwelling longwave radiation $\mathrm{LW}^{\uparrow}$ is described by the Stefan-Boltzmann law. The latent $H_{\mathrm{L}}$ and sensible $H_{\mathrm{S}}$ heat fluxes are estimated by the respective bulk approach (e.g. Gill, 1982). The residual heat flux $Q_{\mathrm{M/R}}$ is calculated from the difference of melting $M$ and refreezing $R$ and keeps track of any heat flux surplus or deficit to keep the ice surface temperature $T_{\mathrm{s}}$ below or equal to 0°C over snow and ice.

The surface mass balance (SMB) in SEMIC is considered as follows

$$\mathrm{SMB} = P_{\mathrm{s}} - SU - M - R \,, \tag{2}$$

where $P_{\mathrm{s}}$ is the rate of snowfall and $SU$ the sublimation rate, which is directly related to the latent heat flux. The melt rate $M$ is dependent on the snow height; if all snow has melted down, the excess energy is used to melt the underlying ice. The refreezing $R$ is calculated differently for available melt water or rainfall. Moreover, the porous snowpack could retain a limited amount of meltwater, while over ice surfaces refreezing is neglected and all melted ice is treated as run-off. In SEMIC, the total melt rate and refreezing rate are calculated from the available energy during the course of one day. As the set of equations is solved using an explicit time-step scheme with a time step of one day, a parameterization for the diurnal cycle (a cosine function) account for thawing and freezing over a day. This reduces complexity, the one-layer snowpack model saves computational time and allows for integration on multi-millennial timescales as opposed to more sophisticated multilayer snowpack models. Further details are given by Krapp et al. (2017).

## 2.2 Atmospheric forcing

Here we targeted peak and decline scenarios in particular, temporarily exceeding a given temperature limit of global warming to 2.0°C or even 1.5°C by the end of 2100. From the official extended RCP2.6 scenarios (Meinshausen et al., 2011), we have selected GCMs which cover the CMIP5 historical scenario, the RCP2.6 scenario until 2299 and reveal an overshoot in annual global mean near-surface temperature change relative to pre-industrial levels (1661–1860). Three different GCMs were used in our study: IPSL-CM5A-LR (L'Institut Pierre-Simon Laplace Coupled Model, version 5 (low resolution)), MIROC5 (Model for Interdisciplinary Research on Climate, version 5) and HadGEM2-ES (Hadley Centre Global Environmental Model 2, Earth System). Instead of the full acronyms we use IPSL, HadGEM2 and MIROC5 in the following text. The GCM output was provided and prepared by the ISIMIP2b project following a strict simulation protocol (Frieler et al., 2017). Figure 1a displays the temporal evolution of the annual global mean near-surface air temperature $T_{\mathrm{a}}$ for those GCMs for the historical simulation up to 2005 continued with the RCP2.6 simulation up to 2299. Global-mean-temperature projections from IPSL, HadGEM2

and MIROC5 under RCP2.6 exceed 1.5°C relative to pre-industrial levels in the second half of the 21[st] century. While global-mean-temperature change returns to 1.5°C or even slightly lower by 2299 in HadGEM2, it only reaches about 2°C in IPSL by 2299. For MIROC5, temperature stabilize at about 1.5°C during the second half of the 21[st] century. In order to determine the onset of overshoot we scan the historical and RCP2.6 scenarios of the individual GCMs identifying the time when the global warming reaches 1.5°C in a 30-year moving window above pre-industrial levels. The characteristic date of overshooting 1.5°C for HadGEM2 is by 2023; MIROC5 reaches this level by 2043, while IPSL reaches this point by 2009 (colored dots in Fig. 1).

The phenomenon that tends to produce a larger change in temperature near the poles was termed polar amplification. Particularly, it enhances the increase in global mean air temperature over arctic areas (referred here as arctic amplification). Generally, the CMIP5 models show an annual average warming factor over the Arctic between 2.2 and 2.4 times the global average warming (IPCC, 2013, Tab. 12.2). As mechanisms creating the arctic amplification may be represented to different extents in the GCMs, the level of future amplification is different across the GrIS. The three GCMs used in this study represent this trend to differing extents over GrIS[1] (Fig. 1 and 2). For HadGEM2 and IPSL the arctic compared to the global warming is amplified relatively similar (warming approximately 4°C relative to 1661–1860). In contrast, MIROC5 reveals a considerably lower arctic amplification (warming approximately 3°C relative to 1661–1860). In terms of global and arctic future annual mean near-surface temperatures MIROC5 offers the lowest and IPSL the highest forcing.

The ISIMIP2b atmospheric forcing data is CMIP5 climate model output data that has been spatially interpolated to a regular $0.5° \times 0.5°$ latitude-longitude grid and bias-corrected using the observational dataset EWEMBI (Frieler et al., 2017; Lange, 2017). To drive the SEMIC model, we need to provide the atmospheric forcing (consisting of incoming shortwave radiation $SW^{\downarrow}$, longwave radiation $LW^{\downarrow}$, near-surface air temperature $T_a$, surface wind speed $u_s$, near-surface specific humidity $q_a$, surface air pressure $p_s$, snowfall rate $P_s$, and rainfall rate $P_r$). These fields are available from the three GCMs model output data. SEMIC is driven by the daily input of the GCMs while the output is the cumulative surface mass balance and the mean surface temperature over each year.

Given the differences in resolution between the GCMs and ISSM, a vertical downscaling procedure is applied to the atmospheric forcing fields. First, the atmospheric fields are interpolated bilinearly from the GCM grid onto a regular high resolution $0.05°$ grid on which SEMIC is run. As a result, the output fields of SEMIC are conservatively interpolated onto the unstructured ISSM grid. This two-step procedure is not necessary, but currently the easiest way from a technical standpoint. For future applications we will avoid the intermediate interpolation and run SEMIC directly on the unstructured target ISSM grid. To account for the difference in ice sheet surface topography between GCMs and ISSM, corrections for several quantities $(\cdot)$ denoted by $(\cdot)^{cor}$ are initially performed. We follow the corrections proposed by Vizcaíno et al. (2010)

$$(\cdot)^{cor} = (h_s^{SEMIC} - h_s^{GCM})\gamma_{(\cdot)}, \tag{3}$$

with the lapse rates $\gamma_{(\cdot)}$ shown in Tab. 1 and $h_s^{SEMIC}$ equal to the ISSM ice surface elevation at the initial state. Subsequently, SEMIC computes the ice-surface temperature $T_s$ and the surface mass balance SMB based on these corrected input values.

---

[1]For all occurrences, the area of the GrIS is defined as the ice mask provided from BedMachine Greenland (Morlighem et al., 2014).

**Table 1.** Lapse rates and height-desertification relationship for initial corrections of GCM output fields near-surface air temperature $T_a$, precipitation of snow $P_s$, precipitation of rain $P_r$, and downwelling longwave radiation $LW^\downarrow$ used as input for SEMIC. Here, $h^{\text{ref}} = 2000\,\text{m}$ is the reference height and $\gamma_p = -0.6931\,\text{km}^{-1}$ is the desertification coefficient.

| variable | lapse rates $\gamma$ and desertification relationship | reference |
|---|---|---|
| $T_a$ | 0.74 K/100 m | Erokhina et al. (2017) |
| $LW^\downarrow$ | 2.9 W m$^{-2}$ | Vizcaíno et al. (2010) |
| $P_s, P_r$ | $\exp(\gamma_p[\max(h_s^{\text{SEMIC}}, h^{\text{ref}}) - h^{\text{ref}}]) \quad \forall \quad h_s^{\text{GCM}} \leq h^{\text{ref}}$ | Vizcaíno et al. (2010) |
| $P_s, P_r$ | $\exp(\gamma_p[\max(h_s^{\text{SEMIC}}, h^{\text{ref}}) - h_s^{\text{GCM}}]) \quad \forall \quad h_s^{\text{GCM}} > h^{\text{ref}}$ | Vizcaíno et al. (2010) |

SEMIC is applied as developed by Krapp et al. (2017). These authors perform a particle-swarm optimization to calibrate model parameters for the GrIS and validate them against the RCM MAR. We adopt their derived parameters here. The parameter tuning aimed to find a parameter set which gives the best fit between SMB and ice temperature $T_s$ of SEMIC with only a limited number of processes and simpler parameterizations compared to a more complex RCM. An RCM is typically validated against reanalysis data and observations, therefore, we assume the tuned parameters are most reliable to represent the processes and parameterizations within SEMIC. In terms of process description, the optimized SEMIC configuration leads to the best possible SMB and $T_s$ fields when MAR is used as forcing. If SEMIC is tuned with another RCM (e.g. RACMO or HIRHAM), the parameters will be different. Here, a separate tuning for each GCM would be required due to the differences (e.g. the timing of maximum warming, the length of an overshoot) among the GCMs used in this study. This basically means to compensate, for e.g. too low near surface temperatures, with SEMIC parameters, which would offset the whole comparison of GCM forcing. Furthermore, these additional tuning steps would make the benefit of having a semi-complexity model with low costs meaningless.

Figure 3 compares averaged SMB fields for the time period 1960–1990 from RACMO2.3 and, exemplary, the SMB derived by forcing SEMIC with HadGEM2. The pattern of the SMB derived by forcing SEMIC with IPSL or MIROC5 is the same as when using HadGEM2. The comparison in Fig. 3 shows that the large-scale pattern of the SMB fields are in fairly good agreement while the small-scale pattern and magnitudes of the GCM-based SMB do not match the RACMO2.3 SMB. Although the coarse GCM-based forcing has undergone a downscaling of particular fields and is processed in SEMIC with a higher resolution, the atmospheric fields over the ice sheet still lack details compared to a RCM. This is due to the fact, that the forcing from a GCM implies different characteristics, like smoother gradients and a less resolved geometry compared to the RCM. The direct output of the SMB from SEMIC to the RACMO2.3 field has a misfit of about $2\,\text{m a}^{-1}$ and a correlation coefficient of $R^2=0.5$. Additionally, the spatially-integrated SMB for the averaged time period differs by up to $200\,\text{Gt a}^{-1}$. For HadGEM2, IPSL and MIROC5 the values are 536, 496, and $614\,\text{Gt a}^{-1}$, respectively. In contrast, the corresponding value for RACMO2.3 is $403\,\text{Gt a}^{-1}$. Therefore, we conclude that the absolute fields from SEMIC are not ideal for our purpose.

Instead of using the absolute SEMIC output fields (SMB and $T_s$) directly to force the numerical ice flow model ISSM, we rely on an anomaly method. The climatic boundary conditions applied here consist of a reference field onto which climate

change anomalies from SEMIC are superimposed. The initialization of the ice flow model based on data assimilation (Sect. 2.6 below) makes it possible to use forcing data from high resolution RCMs that were run on the same ice sheet mask and ice surface topography. As the reference SMB field we choose the downscaled RACMO2.3 product (Noël et al., 2018) whereby a model output was averaged for the time period 1960–1990, denoted $\overline{\text{SMB}}(1960-1990)_{\text{RACMO}}$. The reference period 1960–1990 is chosen as the ice sheet is assumed close to steady state in this period (e.g. Ettema et al., 2009). The climatic SMB that is used as future climate forcing reads

$$\text{SMB}_{\text{clim}}(x,y,t) = \overline{\text{SMB}}_{\text{RACMO}}^{(1960-1990)}(x,y) + \Delta\text{SMB}(x,y,t), \tag{4}$$

with the anomaly defined as

$$\Delta\text{SMB}(x,y,t) = \text{SMB}_{\text{SEMIC}}(x,y,t) - \overline{\text{SMB}}_{\text{SEMIC}}^{(1960-1990)}(x,y), \tag{5}$$

where $t=\{1960, 1961, ... , 2299\}$. Note that the historical scenario is run from 1960–2005 and followed by the RCP2.6 scenario from 2006–2299. In an ideal case, both reference terms $\overline{\text{SMB}}(1960-1990)_{\text{RACMO}}$ and $\overline{\text{SMB}}(1960-1990)_{\text{SEMIC}}$ will cancel out and the absolute climatic forcing $\text{SMB}_{\text{SEMIC}}(x,y,t)$ would remain. This is certainly not the case and the equation must be interpreted as having the RACMO2.3 reference field (with a good spatial distribution) as a background field with the trends from SEMIC superimposed.

The same equations hold for the temperature imposed on the ice-surface. This ensures that the unforced control experiment produces identical behavior for each GCM. Results for future projections depend only on the atmospheric GCM input, or similarly SEMIC output, and therefore the results can be compared quantitatively. In the following text, the constructed SMB fields according to Eq. 4 are referred to as SEMIC-HadGEM2, SEMIC-IPSL and SEMIC-MIROC5 or in general as SEMIC-GCM.

In the presented study, the ice flow model is forced with the off-line processed SEMIC output. As the ice sheet evolves in response to climate change, local climate feedback processes are not captured. Most importantly this includes the interaction of the ice surface between air temperature and precipitation, which in turn affects the surface mass balance. The SMB-feedback process is considered with a dynamic correction to the $\text{SMB}_{\text{clim}}$ (see sect. 2.7 below). This correction is applied within ISSM and to the surface mass balance term only.

## 2.3 Modified RCP2.6 scenario without overshoot

The global climate warming of the selected GCMs exceeds the political target of $1.5°C$ during the $20^{\text{th}}$ century, although the RCP2.6 is the strongest mitigation scenario (Moss et al., 2010). In order to estimate the effect of overshooting on the projected sea-level contribution from the GrIS we manually construct a RCP2.6-like scenario without an overshoot, assuming an immediate climate stabilization at that time when $1.5°C$ is reached. The characteristic time of overshooting $1.5°C$ for HadGEM2 is by 2023; MIROC5 reaches this level by 2043, while IPSL reaches this point by 2009. Before reaching this threshold, the unaltered historical and RCP2.6 forcing is applied. The extension of the forcing from these characteristic times is of crucial importance. To avoid an unphysical step change, the climate in the repeated time period should stabilize (i.e. no

long term trends in temperature change) close to 1.5°C warming. In order to account for decadal variability, i.e. extreme years, we reuse the climatic forcing fields from 2250–2280 until the end of the simulation (light gray shaded areas in Fig. 1 and 4). In this time window, the warming is close to 1.5°C and exhibits a frequent number of extreme years. Other time windows might also be feasible (e.g. the last 30 or 50 years), but will likely not change the forcing substantially. In the following, the modified

RCP2.6-like scenario without overshoot is termed as RCP2.6 without overshoot.

## 2.4 Assessment of SMB forcing

We want to emphasize that we do not intend to validate the energy balance model SEMIC itself, but assess if the obtained SMB fields by forcing SEMIC with the GCMs are plausible. In order to do so, the obtained climatic $SMB_{clim}$ (Eq. 4), the resulting SMB patterns and time series are compared with other available data-sets. Beside the spatial pattern of the surface

mass balance, the time series of the SMB over Greenland illustrates what the ice sheet's total surface gains and losses have been. The constructed SMB forcing for the RCP2.6 scenario with and without overshoot are shown in Fig. 4a and b, respectively. The gray shaded box and black line depict the range and the mean SMB between 1981–2010 from polarportal (polarportal.dk), derived from a combination of observations and a weather model for Greenland (Hirlam-Newsnow). The dashed black line shows the results from the RACMO2.3 product. The spatially-integrated SMB magnitude of each SEMIC-GCM is consistent

with the RACMO2.3 and polarportal data. The drop in SMB after 2000 is present in all three SEMIC-GCMs and RACMO2.3.

For SEMIC-HadGEM2 the spatially-integrated SMB remains around $200\,\mathrm{Gt\,a^{-1}}$ after 2050. The SMB for SEMIC-IPSL recovers from 2050 onwards and shows an increase from around $200\,\mathrm{Gt\,a^{-1}}$ to around $350\,\mathrm{Gt\,a^{-1}}$ by 2300. SEMIC-MIROC5 reveals the lowest SMB change over time and recovers after 2050 from $250\,\mathrm{Gt\,a^{-1}}$ to $300$–$350\,\mathrm{Gt\,a^{-1}}$ by 2300. By 2300, the SMB of SEMIC-IPSL and SEMIC-MIROC5 is slightly below the present-day magnitude. However, the decline of SMB for

the RCP2.6 scenario roughly corresponds with MAR results forced with the GCM NorESM1-M under the RCP2.6 scenario (Fettweis et al. (2013) and last column in Tab. 2), although it is not strictly comparable because they use different GCM climate data. They estimated a loss of -124±100 $\mathrm{Gt\,a^{-1}}$ in 2080–2099 relative to 1980–1999.

Table 2 shows annual mean integrated SMB over the entire GrIS for various periods. Averaged over most of the periods, the annual mean integrated SMB is rather similar among the models. Most obvious are the differences between the SEMIC-

GCMs for the period 1997–2016. The year 1997 was identified as the critical time of Greenland's peripheral glaciers and ice caps mass balance decrease (Noël et al., 2017). For this period of declining SMB, the SEMIC-HadGEM2 agrees well with the RACMO2.3 product while the spatially-integrated SMB for SEMIC-IPSL and SEMIC-MIROC5 are ∼40 and $50\,\mathrm{Gt\,a^{-1}}$ larger, respectively.

For the available RACMO2.3 time series and the SEMIC-GCMs, we have computed the interannual SMB variability (Fig. 5).

The SMB variability is in terms of frequency and amplitude similar to RACMO2.3 but is not coherent among all models because the GCMs have their own internal variability. For the time period 1960–2016, the overall surface mass balance difference over the ice sheet between SEMIC-GCM and RACMO2.3 is almost zero with -0.007 $\mathrm{m\,a^{-1}}$, 0.016 $\mathrm{m\,a^{-1}}$ and 0.0200 $\mathrm{m\,a^{-1}}$ for SEMIC-HadGEM2, SEMIC-IPSL and SEMIC-MIROC5, respectively. These numbers are in the same range as given by Krapp et al. (2017) for the comparison between SEMIC and MAR.

**Table 2.** Annual mean integrated SMB (Gt yr$^{-1}$) covering various periods. Time series of SMB$_{\text{clim}}$ for the SEMIC-GCMs are calculated by Eq. 4 for RCP2.6 scenario. The column '1.5°C reached' shows a 30-year mean at the characteristic time of overshooting 1.5°C. Anomaly in SMB (ΔSMB) is in 2080–2099 with respect to 1980–1999.

| Model | 1960–1990 | 1960–1997 | 1997–2016 | 1981–2010 | 1960–2016 | 1.5°C reached | ΔSMB |
|---|---|---|---|---|---|---|---|
| RACMO2.3 | 402.8 | 403.4 | 279.1 | 363.1 | 364.8 | - | - |
| polarportal | - | - | - | 370 | - | - | - |
| MAR [a] | - | - | - | - | - | - | −124±100 |
| SEMIC-HadGEM2 | 400.0 | 391.2 | 277.0 | 358.1 | 355.2 | 170.0 | −179.2 |
| SEMIC- IPSL | 408.9 | 412.5 | 332.8 | 403.7 | 382.2 | 363.9 | −170.4 |
| SEMIC-MIROC5 | 395.0 | 398.5 | 341.2 | 341.8 | 380.0 | 288.4 | −80.9 |

[a] MAR forced with GCM NorESM1-M under RCP2.6 scenario (Fettweis et al., 2013)

## 2.5 Ice flow model

Ice flow and thermodynamic evolution of the GrIS are approximated using the finite-element based ISSM. The model has been applied successfully to both large ice sheets (Bindschadler et al., 2013; Nowicki et al., 2013; Goelzer et al., 2018) and is also used for studies of individual drainage basins of Greenland, e.g. the North East Greenland Ice Stream (Choi et al., 2017), Jakobshavn Isbræ (Bondzio et al., 2016, 2017) and Store Glacier (Morlighem et al., 2016). Here, we use an incompressible non-Newtonian constitutive relation with viscosity dependent on temperature, microscopic-water content and strain rate, while neglecting the softening effect of damage or impurities. The BP approximation to the Stokes momentum balance equation is employed in order to account for longitudinal and transverse stress gradients.

ISSM is specified with kinematic boundary conditions at the upper and lower boundary of the ice sheet. The upper boundary incorporates the climatic forcing obtained from SEMIC as explained above, i.e. the surface mass balance and ice surface temperature. The ice surface temperature is prescribed through Dirichlet boundary conditions. The base of grounded ice is specified as both impenetrable with the bedrock and in balance with the rate of basal melting. At the base of floating ice we use a Neumann boundary condition that parameterizes the heat flux at the ice-ocean interface (Eq. 27 in Larour et al., 2012). The basal melt rate below ice shelves is parameterized with a Beckmann-Goosse relationship (Beckmann and Goosse, 2003). The melt-factor is roughly adjusted such that melting rates correspond to literature values (e.g. Wilson et al., 2017). Within this study the basal melt rate is not a focus and hence the basal melt underneath floating tongues or vertical calving fronts of tidewater glaciers are not changed. Once the pressure melting point at the grounded ice is reached, melting is calculated from basal frictional heating and the heat flux difference at the ice/bed interface. At the ice base sliding is allowed everywhere and the basal drag, $\boldsymbol{\tau}_b$, is written using Coulomb friction:

$$\boldsymbol{\tau}_b = -k^2 N \boldsymbol{v}_b, \tag{6}$$

where $\boldsymbol{v}_b$ is the basal velocity vector tangential to the glacier base and $k^2$ a constant. The effective pressure is defined as $N = \varrho_i\, g\, H + \varrho_w\, g\, h_b$, where $H$ is the ice thickness, $h_b$ the glacier base and $\varrho_i = 910\,\mathrm{kg\,m}^{-3}$, $\varrho_w = 1028\,\mathrm{kg\,m}^{-3}$ the densities for ice and sea water, respectively. We apply water pressure at the calving front of marine terminating glaciers and observed surface velocities (Rignot and Mouginot, 2012) at the ice front of land terminating glaciers. A traction-free boundary condition

is imposed at the ice/air interface.

Geothermal heat flows into the ice in contact with bedrock and adjusts dynamically to the thermal state of the base (Aschwanden et al., 2012; Kleiner et al., 2015). The spatial pattern of the geothermal flux is taken from Greve (2005, scenario hf_pmod2).

For all simulations, the ice front is fixed in time, and a minimum ice thickness of 10 m is applied. This implies that calving

and melting exactly compensates the outflow through the margins and initially glaciated points are not allowed to become ice-free. However, regions that reach this minimum thickness have retreated. The grounding line is allowed to evolve freely according to a sub-grid parameterization scheme, which tracks the grounding line position within the element (Seroussi et al., 2014).

Model calculations are performed on a horizontally unstructured grid with a higher resolution, $l_{\min} = 1\,\mathrm{km}$, in fast flow

regions and with a coarser resolution, $l_{\max} = 20\,\mathrm{km}$, in the interior. The vertical discretization comprises 15 layers refined towards the base where vertical shearing becomes more important. The complete mesh comprises 574 056 elements. Velocity, enthalpy (i.e. temperature and microscopic water content) and geometry fields are computed on each vertex of the mesh using piecewise-linear finite elements. The Courant-Friedrichs-Lewy condition (Courant et al., 1928) dictates a time step of 0.025 years. Using the AWI cluster Cray-CS 400 computer, a simulation with an integration time of 340 years requires $\approx 8$

hours on 16 nodes comprised of 36 CPUs.

## 2.6   Initial state

Future projections of ice sheet evolution first require the determination of the initial state. Different methods are currently used to initialize ice sheets and it has been shown, that the initial state is crucial for projections of ice dynamics (Bindschadler et al., 2013; Nowicki et al., 2013; Goelzer et al., 2018). The recent initMIP-GrIS intercomparison effort (Goelzer et al., 2018)

focuses on the different initialization techniques applied in the ice flow modeling community and found none of them is the method of choice in terms of a good match to observations and a long term continuity. All methods are required for modeling the projections of the GrIS planned within CMIP6 phase (Nowicki et al., 2016) on time scales of up to a few hundred years. However, while inverse modeling is well established for estimating basal properties, the temperature field is difficult to constrain without performing an interglacial thermal spin-up.

Here, we employ a hybrid approach between spin-up and an inversion scheme to estimate the initial state. For the hybrid initialization we make the three basic simplifications: (1) The currently observed present-day elevation is taken constant for the entire glacial cycle. (2) the basal friction coefficient obtained from the inversion is taken constant for the past glacial cycle, and (3) the temperature changes from the GRIP record are applied to the whole ice sheet without spatial variations.

The ice sheet geometry (bed, ice thickness and ice sheet mask) is taken from the mass-conserving BedMachine Greenland data set v2 (Morlighem et al., 2014). The geometric input for thickness and ice sheet mask is masked to exclude glaciers and ice caps surrounding the ice sheet proper. An initial relaxation run over 50 years assuming no sliding and a constant ice temperature of -20°C is performed to avoid spurious noise that arises from errors and biases in the datasets. A temperature spin-up is

then performed using this time-invariant geometry. As the computationally expensive BP approximation is employed, mesh refinements are made at certain points during the whole initialization procedure (see Table 3). The first mesh sequence starts 125 kyr before the present day and runs up to the year 1960, assuming a spatially constant friction coefficient $k^2 = 50\,\mathrm{s\,m^{-1}}$ and is forced with paleo-climatic conditions. The imposed paleo-climatic conditions consist of a multi-year mean from the years 1960 to 1990 of the RACMO2 product (Ettema et al., 2009) and offset by a spatially constant surface temperature anomaly

for the last 125 kyr based on the GRIP surface temperature history derived from the $\Delta^{18}O$ record (Dansgaard et al., 1993). The initial ice temperature at 125 kyr before present is a steady-state temperature distribution taken from a spin-up with time independent climatic conditions from the reference period 1960–90. The spin-up is done up to 1960 to start the projections before the critical time of Greenland's peripheral glaciers mass balance decrease (Noël et al., 2017) with an additional buffer of approximately 30 years.

In the subsequent basal-friction inversion, the ice rheology is kept constant using the enthalpy field from the end of the temperature spin-up. The inversion approach infers the basal friction coefficient $k^2$ in Eq. 6 by minimizing a cost function that measures the misfit between observed and modeled horizontal velocities (Morlighem et al., 2010). Observed horizontal surface velocities are taken from Rignot and Mouginot (2012). The cost function is composed of two terms which fit the velocities in fast- and slow-moving areas. A third term is a Tikhonov regularization to avoid oscillations. The parameters for weighting the

three contributions to the cost functions are taken from Seroussi et al. (2013).

The procedure of temperature spin-up and inversion is repeated on the subsequent three mesh sequences. The repeated temperature spin-ups start 125 kyr, 25 kyr and 15 kyr before 1990 and again run up to the year 1960. The initial values for the temperature field at these times are taken from the respective times from the previous mesh sequence; the basal-friction coefficient is updated from the inversion on the previous mesh sequence. The mesh sequencing reduces the expense of initialization

and produces a sufficiently consistent result in terms of velocity and enthalpy. Note that mesh sequence 1-3 is only used during initialization while the final solution of mesh sequence 4 at year 1960 of this procedure is used as initial state for all projections presented below.

Please note, that similar results from this procedure have been submitted to the ISMIP6 initMIP-Greenland effort (Goelzer et al., 2018), but the simulations were run with the geothermal flux distribution by Shapiro and Ritzwoller (2004) and addition-

ally with a time independent climate forcing representing present-day conditions. However, by using the modified heat-flux distribution by Greve (2005, scenario hf_pmod2), we found a generally better agreement with measured basal temperatures at ice core locations. Basically, the comparison of simulated to observed temperatures at the ice base shows too low temperatures for some locations. As the applied inversion technique for the friction coefficient allows sliding everywhere, the portion of deformational shearing may be underestimated, which cannot be proven without any observations of basal velocities, which

**Table 3.** Mesh Statistics.

| mesh sequence | $l_{\min}$ (km) | $l_{\max}$ (km) | number of elements | integration time in thermal spin-up (kyr) |
|:---:|:---:|:---:|:---:|:---:|
| 1 | 15 | 50 | 117 586 | 125 |
| 2 | 5 | 50 | 192 220 | 125 |
| 3 | 2.5 | 35 | 272 650 | 25 |
| 4 | 1 | 20 | 574 056 | 15 |

unfortunately do not exist. However, for our projections on centennial timescales this is a negligible effect (Seroussi et al., 2013).

## 2.7 Synthetic and dynamic surface mass balance parameterization

As we perform a one-way coupling of the climatic forcing, the SMB-elevation feedback needs to be considered. Here we rely on the dynamic SMB parameterization developed by Edwards et al. (2014a, b) and previously applied by Goelzer et al. (2013). This relationship was estimated from a set of MAR simulations in which the ice sheet surface elevation was altered. The parameterization assumes that the effect of SMB trends follow a linear relationship

$$\mathrm{SMB}_{\mathrm{dyn}}(x,y,t) = \mathrm{SMB}_{\mathrm{clim}}(x,y,t) + b_i(h_s(x,y,t) - h_{\mathrm{fix}}(x,y)), \tag{7}$$

where $\mathrm{SMB}_{\mathrm{dyn}}(x,y,t)$ and $\mathrm{SMB}_{\mathrm{fix}}(x,y,t)$ are the SMB values with and without taking height changes into account, respectively. The surface elevation changes are taken from the ISSM elevation $h_s(x,y,t)$ while running the simulation and a reference elevation $h_{\mathrm{fix}}(x,y)$. In our setup the reference elevation corresponds to the ISSM ice surface elevation at the initial state.

In this parameterization the SMB gradient $b_i$ is dependent on both location and sign. It can take four values and a separation is made on the location relative to 77°N and on the sign of the SMB. This separates regions of largely different sensitivity, namely the ablation zone with a larger gradient compared to the accumulation zone, and a more sensitive ablation zone in the south compared to the north. While a complete uncertainty analysis is given by Edwards et al. (2014a), only the maximum likelihood gradient set, $\boldsymbol{b} = (b_p^N, b_n^N, b_p^S, b_n^S)$, is used here:

$$b_p^N = 0.085 \, \mathrm{kg \, m^{-3} \, a^{-1}},$$
$$b_n^N = 0.543 \, \mathrm{kg \, m^{-3} \, a^{-1}},$$
$$b_p^S = 0.063 \, \mathrm{kg \, m^{-3} \, a^{-1}},$$
$$b_n^S = 1.890 \, \mathrm{kg \, m^{-3} \, a^{-1}},$$

where the subscripts $(p,n)$ and the superscripts $(N,S)$ indicate the evaluation of the SMB sign and the region separation, respectively. Please note, that the employed relationship with their parameters may change using a setup from SEMIC.

A shortcoming of the performed hybrid initialization is, that usually a fixed initial ice sheet causes a model drift when imposing the ice thickness equation. This is a result from using an ice sheet that is not in equilibrium with the applied SMB and ice flux divergence. We utilize the local ice thickness imbalance once the ice sheet is released from its fixed topography from a single year unforced relaxation run, i.e. $\Delta\mathrm{SMB}(x,y,t) = 0$ in Eq. 5. The resulting $\partial H/\partial t$ is subtracted as a surface mass balance correction, $\mathrm{SMB}_{\mathrm{corr}}(x,y)$, for all further runs (similar as in Price et al. (2011); Goelzer et al. (2018)). However, instead of assuming a zero SMB anomaly, one could calculate the anomaly with a GCM input from the CMIP5 pre-industrial scenario. But given the small temperature changes, the SMB anomaly will be close to zero and the calculated ice thickness imbalance is unlikely to be affected by it. However, the final SMB correction is on average $0.01\,\mathrm{m\,a^{-1}}$, with 5% of the total ice-sheet area having a correction of $>25\,\mathrm{m\,a^{-1}}$, predominantly at marine-terminated ice margins and ice streams (Fig. 6). For these locations, the synthetic SMB correction can be considered additional ice thinning or thickening from dynamic discharge that is not intrinsically simulated. A performed control run with the imposed SMB correction exhibits a small model drift in terms of sea-level equivalent (SLE, black dashed line in Fig. 11 and section 3.3).

The final surface mass balance that the numerical ice flow model sees is composed of several components

$$\mathrm{SMB} = \mathrm{SMB}_{\mathrm{clim}}(x,y,t) - \mathrm{SMB}_{\mathrm{corr}}(x,y) + \mathrm{SMB}_{\mathrm{dyn}}(x,y,t)\,. \tag{8}$$

# 3 Results

## 3.1 Forcing fields

For the different GCMs used we compute ice surface temperature $T_s$ differences between 2100/2300 and 2000 as a multi-year mean over five years to reduce the inter-annual variability (Fig. 7). HadGEM2 leads to an increase in temperatures along the northern margins of up to $4°C$. By 2100 the western areas and vast majority of the ice sheet exceed $2°C$ of warming. The only pronounced warming by 2300 is in the northwestern regions, while the ice sheet surface temperatures decrease compared to 2100. IPSL exhibits a significantly different pattern with pronounced warming in the center (up to $3°C$) and in the southeast (up to $4°C$) of the ice sheet. The northern areas reveal moderately warming around $1°C$ by 2100. The pattern is similar in 2300, with a moderate cooling in the west compared to 2100. The least warming is found in MIROC5, which even exhibits cooling in the southern areas by about $-1°C$ in 2100; warming of $+1°C$ is only reached in the north. By 2300 the entire ice sheet experiences warming; however this warming is quite moderate compared to the other two GCMs. The low magnitude of warming over Greenland compared to global warming let us infer that the mechanisms of arctic amplification is not well represented in MIROC5.

Although we do not have a measure to judge future climate warming trends, with respect to the Arctic amplification phenomena the most plausible distribution and magnitude of surface warming is produced by HadGEM2. By contrast, MIROC5 produces less pronounced warming over Greenland that is similar to the global mean warming but exhibits a plausible pattern of warming. IPSL is spatially and temporally experiencing the largest warming; however, the distribution is not in agreement with the Arctic amplification. Still, the assessment of the GCMs is in line with skill tests performed by Watterson et al. (2014)

on a global scale. They assigned skill cores by comparing individual GCM output data against re-analysis data. The analysis indicates that all 25 models have a substantial degree of skill, however, HadGEM2 is ranked in the top, MIROC5 in the middle, and IPSL in the lower part.

Figure 8 presents, in a similar fashion as Fig. 7, the differences in SMB between 2100/2300 and 2000 as a multi-year mean over five years each. The difference in SMB 2100-2000 of SEMIC-HadGEM2 indicates a similar pattern as presented by Krapp et al. (2017) using MAR (Fettweis et al., 2013). Increasing SMB in the eastern part of the ice sheet with a maximum in the southern half of the ice sheet; at the ice sheet margins ablation is increased. The same pattern is characteristic for 2300-2000, but with a slight decrease in melting and accumulation. The SMB is reduced in the center, leaving a wide area with differences in SMB of $0.5\,\mathrm{m\,a^{-1}}$ and less. The SMB difference of SEMIC-IPSL is showing a similar pattern with enhanced amplitudes compared to SEMIC-HadGEM2, in particular, and the southwestern margin; melting in the southwest is increased by up to $1\,\mathrm{m\,a^{-1}}$. In contrast, an SMB gain is concentrated in the center-east by 2300. The most astonishing result is the $\Delta$SMB pattern in SEMIC-MIROC5. Increasing the SMB along the southwestern and southern margins in contrast to gently decreasing the SMB in the center of the ice sheet. By 2300 $\Delta$SMB the pattern changes slightly and the SMB is decreasing in the southwestern margins. The magnitude of $\Delta$SMB is lower compared to SEMIC-HadGEM2 and SEMIC-IPSL.

## 3.2 Present day elevation and velocities

Figure 9 displays, exemplified, the observed and simulated velocities for the year 2000 (defined here as present day) after a period of forcing with SEMIC-HadGEM2 from 1960 onwards. The resulting horizontal velocity field captures all major features well, including the North East Greenland Ice Stream (NEGIS). Outlet glaciers terminating in narrow fjords in the southeastern region are resolved, however, slow moving areas tend to retreat below minimum ice thickness and with that the ice extent in this area is underestimated. However, ice surface elevations agree fairly well (Fig. 10b). In general large outlet glaciers like Kangerdlusuaq, Helheim and Jakobshavn Isbræ reveal lower velocities in their fast termini that reflects the high RMS of about $400\,\mathrm{m\,a^{-1}}$ (Fig. 10a). The RMS analysis here was done on the native grid with the high resolution in fast flow regions and the model was already run 40 years forward in time. Compared to these values, the AWI-ISSM results on the regular $5\,\mathrm{km}$ grid given in Goelzer et al. (2018) have a lower RMS value of $<20\,\mathrm{m\,a^{-1}}$.

## 3.3 Projections of mass change

After passing the assumed critical time of declining SMB of the GrIS and the present day state, the ice sheet experiences a warming and associated mass loss from a decline in surface mass balance. Projections of the evolution of SLE of the ice sheet under RCP2.6 scenario until 2100 and 2300 are shown in Fig. 11 for each GCM (solid lines) and Table 4. The simulated volume above floatation is converted into the total amount of global sea-level equivalent (SLE) by assuming an ocean area of about $3.618{\times}10^8\,\mathrm{km^2}$. Although the control run shows a small model drift in terms of SLE (-1.4 and -0.7 mm for 2100 and 2300, respectively), the RCP2.6 projected SLE is corrected by the control run. By 2100, the model range of Greenland sea-level contributions is between 21.3 and 38.1 mm with an average of 27.9 mm and by 2300 between 36.2 and 85.1 mm with an average of 53.7 mm. Compared to Fürst et al. (2015) our mean values are lower but still in their model range.

The evolution of the mass change, expressed as sea-level equivalent, (Fig 11) is showing distinct behaviors: between 1960–2000 almost no change for SEMIC-HadGEM2 and SEMIC-IPSL while SEMIC-MIROC5 is gaining mass; a change in trend with a minor increase between 2000–2015 and a steep increase from then on for SEMIC-HadGEM2 and SEMIC-IPSL; SLE increase for SEMIC-MIROC5 is more gentle. The steep rise in SLE for SEMIC-HadGEM2 and SEMIC-IPSL is linked to the steep reduction in SMB for both models at the same time. The kink of SLE in SEMIC-HadGEM2 and SEMIC-IPSL around 2050 is caused by a positive SMB anomaly (compare Fig. 4). Also SEMIC-MIROC5 shows this peak in SMB, however slightly later, around 2060. These short-term drops in SLE are linked to positive anomalies in SMB. For SEMIC-HadGEM2 the ice sheet contribution until 2300 generally increases continuously while for SEMIC-IPSL and SEMIC-MIROC5 the increase levels off. This is an intriguing effect as SEMIC-HadGEM2 and IPSL are showing a similar behavior in terms of warming over GrIS (Fig. 1). In fact, the SMB of SEMIC-IPSL recovers from 2050 onwards (Fig. 4), while the SMB of SEMIC-HadGEM2 remains on a low level.

For the RCP2.6 scenario without overshoot the behavior of SLE for SEMIC-HadGME2 is similar but with lower values. The SLE for SEMIC-MIROC5 is approximately 5 mm lower by 2100 but approaches the same value at 2300 without attaining a pronounced plateau. A striking feature is the much lower SLE estimated from SEMIC-IPSL which never exceeds a value of 10 mm and gains mass from about 2225 onwards. The average SLE from all three GCMs is 17.4 mm by 2100 and 37.1 mm by 2300, that is approximately one third less compared to the RCP2.6 scenario.

The observed sea-level contribution between 2002 and 2014 is $0.73\,\mathrm{mm\,a}^{-1}$ (Rietbroek et al., 2016). In the same period, the simulated contribution is only $0.16\,\mathrm{mm\,a}^{-1}$ for SEMIC-HadGEM2, $0.17\,\mathrm{mm\,a}^{-1}$ for SEMIC-IPSL and lowest for SEMIC-MIROC5 with $0.13\,\mathrm{mm\,a}^{-1}$. In order to assess a potential temporal lag between simulated and observed value, mean values of similar periods are calculated (Fig. 12). None of the models reach the observed value (solid black line in Fig. 12); HadGEM2 reaches a maximum value of $0.59\,\mathrm{mm\,a}^{-1}$ 13 years later; SEMIC-IPSL a value of $0.48\,\mathrm{mm\,a}^{-1}$ 12 years later and SEMIC-MIROC5 a value of $0.36\,\mathrm{mm\,a}^{-1}$ 40 years later. For the RCP2.6 scenario without overshoot, the values are smaller. Since a future ocean forcing and calving front retreat is not considered here, the response of the ice sheet is likely underestimated. Comparing the sea-level contributions of each SEMIC-GCM to the sea-level contribution of $0.4\,\mathrm{mm\,a}^{-1}$ calculated from RACMO2.3 for the same period (dashed black line in Fig. 12) reveals a better agreement. SEMIC-HadGEM2 reaches this value 8 years later for the RCP2.6 scenario with overshoot and 9 years later for the RCP2.6 scenario without overshoot; SEMIC-IPSL reaches this value 10 years later for RCP2.6 with overshoot.

### 3.4 Ice thickness change and dynamic response

Extensive marginal thinning is experienced by forcing the ice sheet with SEMIC-HadGEM2 and SEMIC-IPSL (Fig. 13). In contrast to the mass loss near the margin the interior shows thickening; IPSL reveals more thickening in the interior. Generally the large-scale pattern of marginal thinning and central thickening correlates with observations (Helm et al., 2014) except that Petermann and Kangerdlusuaq glaciers show an opposite trend. With a forcing of MIROC5 the pattern of the elevation change is different with thinning in the southern center of the ice sheet; the northern center experienced thickening. Although thinning occurs at the margin it is less extensive compared to the other GCMs.

**Table 4.** Contribution of the Greenland ice sheet to global sea-level change by 2100 and 2300 in mm SLE under RCP2.6 scenario with and without overshoot.

| Model / Study | 2100 | | 2300 | |
| --- | --- | --- | --- | --- |
| | with overshoot | without overshoot | with overshoot | without overshoot |
| SEMIC-HadGEM2 | 38.1 | 29.6 | 85.1 | 66.9 |
| SEMIC-IPSL | 24.4 | 7.5 | 36.2 | 3.4 |
| SEMIC-MIROC5 | 21.3 | 15.0 | 39.9 | 40.9 |
| Average | 27.9 | 17.4 | 53.7 | 37.1 |
| Fürst et al. (2015) | 42.3±18.0 | - | 88.2±44.8 | - |

The response of ice velocities to RCP2.6 forcing is presented in Fig. 14, where the change in horizontal surface velocities is shown for all scenarios as a difference between 2100–2000 and 2300–2000 (each as five year mean). For all SEMIC-GCM forcings the ice response shows a fairly similar behavior. The NEGIS, Jakobshavn Isbræ, Helheim, Ryder glaciers and Hagen Bræ experience acceleration; deceleration is present at Petermann and Kangerdlusuaq glaciers. However, the magnitude of response is different across all models. Most prominent at the western margin where SEMIC-HadGEM2 leads to the strongest acceleration while SEMIC-MIROC5 to the lowest.

## 4 Discussion

Fürst et al. (2015) performed a comprehensive ensemble study for a suite of 10 GCMs (HadGEM2-ES, IPSL-CM5A-LR and MIROC5 included) and four different RCP scenarios. For the RCP2.6 scenario they estimate a sea-level contribution of 42.3±18.0 mm by 2100 and 88.2±44.8 mm by 2300. Our averaged result of a sea-level contribution under RCP2.6 forcing is slightly lower but still in their ensemble variability. The resultant projection by Fürst et al. (2015) included contributions from lubrication, marine melt and SMB-coupling while ours accounts for SMB forcing only. The lubrication effect was diagnosed to have a negligible effect on the overall mass budget, but the oceanic influence on the total ice loss explains about half of the mass loss for RCP2.6. Since a future ocean forcing and calving front retreat is not considered here, the response of the ice sheet is likely underestimated here. By 2010 the cumulative ice discharge anomaly for SEMIC-HadGEM2 contributes by about 15% to the ice loss. By 2100 and 2300 the contribution is below 3 and 7% respectively, and becomes negligible. For SEMIC-IPSL and SEMIC-MIROC5 the cumulative effect of ice discharge anomaly shares less than 10% of the total mass budget by 2010 and 2100 but increases towards 17% by 2300. The different behavior can be explained by the interaction with the SMB and ice dynamics as the relative importance of outlet glacier dynamics decreases with increasing surface melt (Goelzer et al., 2013; Fürst et al., 2015). Increased ice discharge causes dynamic thinning further upstream, lowering of the ice surface and thereby intensifies surface melting due to the associated warming of the near surface. Surface melting in turn competes with the discharge increase by removing ice before it reaches the marine margin. The simulated increase of ice discharge for SEMIC-

IPSL and SEMIC-MIROC5 is therefore linked to the recovery of SMB over the course of the 22$^{\text{nd}}$ century. Still, the SMB remains the dominant factor for mass loss. The speed-up observed from all scenarios merely transports ice from the interior but is melted before it reaches the ice sheet margin. However, the values for sea-level contribution of this study may serve as a lower bound, as processes (ocean forcing and calving) proven to play a major role in GrIS mass loss are not yet represented by the model.

Additionally, the calculation of the surface mass balance is based on different methods. Fürst et al. (2015) rely on the rather simple and empirically derived PDD scheme, while we use a more advanced energy-balance approach. So far the sensitivity of melting to warming of this class of models is not well understood. Comparisons of PDD models and energy-balance models suggested that the former are too sensitive to climate change and produce a larger runoff response (van de Wal, 1996; Bougamont et al., 2007; Graversen et al., 2011). On the other hand Goelzer et al. (2013) attempted to make a robust comparison and find that a PDD model underestimates sea-level rise by 14–31% compared to MAR. An Assessment of the SMB and its impact on sea-level contribution calculated by the PDD scheme in Fürst et al. (2015) and the SEMIC model from this study cannot be drawn, because of the strong interaction between ice loss, ice dynamics and external forcings. As the cumulative discharge rates in the mass budget are higher compared to Fürst et al. (2015), this may indicate a lower SMB forcing. However, compared to other models that participate in the initMIP-GrIS exercise (Goelzer et al., 2018), our setup is neither on the higher nor the lower spectrum of estimated mass loss. Additionally, we have conducted SeaRISE experiments similar to Bindschadler et al. (2013), which showed us that we are within the spread among the models, in particular, for the amplified climatic scenarios C1, C2, and C3 (not shown here).

The modified RCP2.6 scenario without overshoot projected a sea-level contribution that is on average about 38% and 31% less by 2100 and by 2300, respectively. For SEMIC-HadGEM2 and SEMIC-MIROC5 the partition of the mass budget is relatively similar to the RCP2.6 scenario but with a slightly increased cumulative discharge anomaly. For SEMIC-IPSL the behavior is more irregular where the ice sheet gains mass during the last century, as a result from an increasing SMB which is partly compensated by enhanced ice discharge of up to 40%. However, the spread of sea-level contribution is much larger compared to the RCP2.6 scenario. In particular, in 2300 the range of sea-level contribution is between 3.4–66.9 mm. The very low estimated contribution of 3.4 mm is a result from the SEMIC-IPSL forcing that predicts a relatively high SMB of 364 Gt yr$^{-1}$ for the characteristic time of overshooting 1.5°C (Column '1.5°C reached' in Tab. 2). The SMB is close to present-day and therefore SEMIC-IPSL maintains a geometry close to the present day. In contrast, SEMIC-HadGEM2 has declined to 170 Gt yr$^{-1}$ and SEMIC-MIROC5 to 288 Gt yr$^{-1}$. The prolongation of these scenarios was done by repeating the forcing from a time window that reveals a stabilized climate. Repeating the last 30-year forcing field window before the characteristic time is not reasonable, because the change in warming is strongest during that period and a stabilized climate would not be reached. In fact, we would generate a non-mitigation pathway scenario with constant warming rates that would have a larger melt and therefore makes up a larger part of the sea-level contribution (not shown here).

The generally abated sea-level contribution is in agreement with the inferred threshold in global mean temperature before irreversible ice sheet topography changes occur. The simplified assumption behind this threshold is an integrated SMB over the whole ice sheet that becomes negative (Gregory and Huybrechts, 2006). Fettweis et al. (2013) reported a threshold of 3.5°C

relative to pre-industrial levels, which is never exceeded under the RCP2.6 scenario. Assuming a steady state ice-sheet SMB of $400\,\mathrm{Gt\,yr^{-1}}$ within the reference period, the decline in SMB must be larger than $-400\,\mathrm{Gt\,yr^{-1}}$ to get a continuous, retreating ice sheet margin. If the mean SMB of the GrIS remains positive, a new steady state ice sheet geometry may be possible, but would require a balancing with the ice outflow.

At last we want to discuss if studying RCP2.6 allows to draw significant conclusions on the development of sea-level rise due to mass loss in Greenland. We found that only a fraction of the current observed mass loss in the first two decades is represented by the model in RCP2.6. This can be attributed to different factors: the current emissions are above the RCP2.6 limit and hence the natural system evolves on a different route than RCP2.6. Secondly, the three GCMs are quite different in response to the RCP2.6 forcing and the ISM used itself does not represent all mechanisms, in particular the lack of oceanic forcing is causing a reduced sea-level rise. Hence, a new emission scenario, that represent the real RCP pathway in the recent past, would be most useful for future studies like ours.

## 5    Conclusions

We have applied climate forcings based on the low-emission scenario CMIP5 RCP2.6 of three underlying GCMs (HadGEM2-ES, IPSL-CM5A-LR, MIROC5) to ISSM. Despite all three GCMs being based on RCP2.6, their temperature variation – globally and regionally for GrIS – is different. Arctic amplification causes a near-surface air temperature increase over Greenland by a factor of $\approx 2.4$ and 2 in HadGEM2-ES and IPSL-CM5A-LR, respectively. MIROC5 reveals nearly no arctic amplification. In order to force the ice sheet model with a reliable SMB, a physically based surface energy balance model of intermediate complexity (SEMIC) was applied. The estimated sea-level contribution for the RCP2.6 peak and decline scenario in our simulations ranges between 21–38 mm by 2100 and 36–85 mm by 2300 and up to 30–40% higher compared to a scenario without overshoot. Despite the reduced SMB in the warmer climate, a future steady-state ice sheet with lower surface and volume might be possible.

Although the thickness change pattern agrees well with observations and acceleration of NEGIS, Helheim Glacier and Jakobshavn Isbræ is captured in our simulations, the estimated sea-level contribution is potentially underestimated due to the following drawbacks of our study: (i) retreat of glaciers due to oceanic forcing (melt at vertical cliffs and/or calving rates) and (ii) seasonality due to lubrication arising from supra-glacial melt water is not included. This leads to the conclusion that the projections may serve as a lower bound of the contribution of Greenland to sea-level rise under the RCP2.6 climate scenario. This limits also the advantageous treatment of the physics in our model setup, meaning that all the benefits from a high-resolution higher order model are not yet contributing to the extent they potentially could. Our results further indicate, that uncertainties stem from the underlying climate model to calculate the surface mass balance.

*Code availability.* The ice sheet model ISSM is available at issm.jpl.nasa.gov and not distributed by the authors of this manuscript. SEMIC is available from https://gitlab.pik-potsdam.de/krapp/semic-project and not distributed by the authors of this manuscript.

*Author contributions.* M.R. conducted ISSM simulations, coupled SEMIC output to ISSM. M.R. and A.H. designed the study, analyzed the results and wrote major parts of the manuscript. K.F. and S.L. selected, prepared and contributed GCM forcings. U.F. has contributed advice on the albedo scheme and checked the GCM input data.

*Competing interests.* There are no competing interests present.

5   *Acknowledgements.* This work was funded by BMBF under grant EP-GrIS (01LS1603A) and the Helmholtz Alliance Climate Initiative (REKLIM). We would like to thank an anonymous reviewer and Clemens Schannwell for the detailed review containing many helpful remarks and constructive criticism that helps to improve the manuscript. We acknowledge the technical support given by Mario Krapp (PIK) with SEMIC. We are grateful for the NetCDF interface to SEMIC provided by Paul Gierz (AWI). We would like to thank Vadym Aizinger, Natalja Rakowsky and Malte Thoma for maintaining excellent computing facilities at AWI. We also enthusiastically acknowledge the general 10   support of the ISSM team.

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

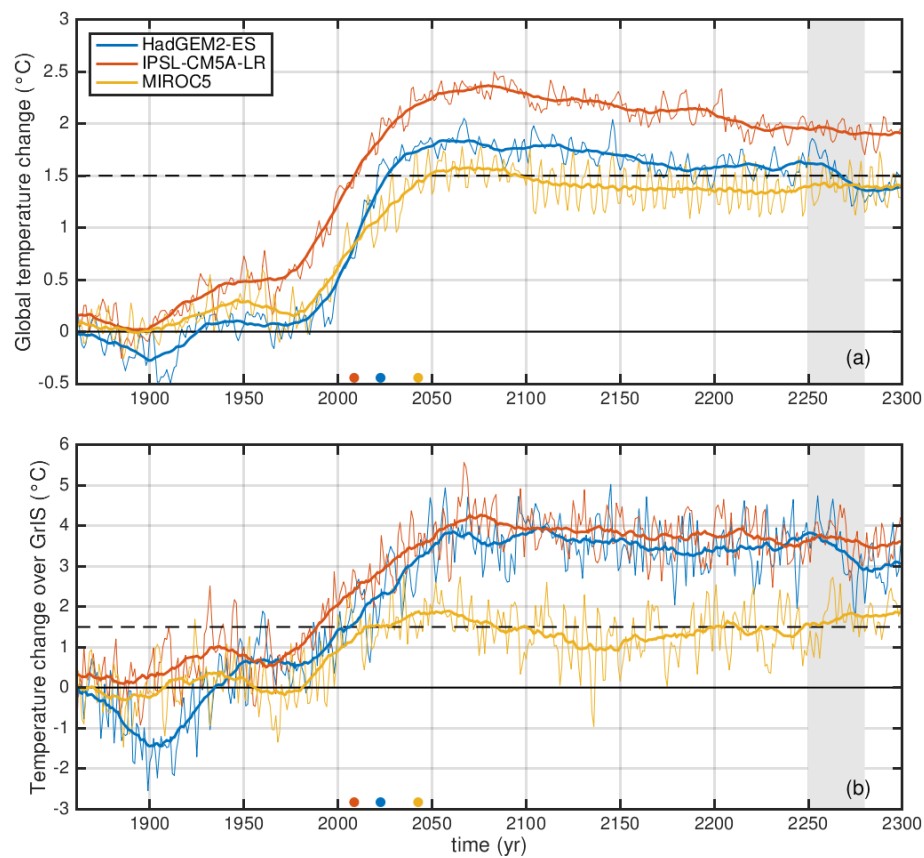

**Figure 1.** Time series of annual global mean near-surface temperature change (a) and over the GrIS (b) for all three GCMs relative to 1661–1880. The thick line is a 30-year moving mean. The colored dots represent the onset years of overshooting 1.5°C in the global mean near-surface air temperature in a 30-year moving window relative to pre-industrial levels. The light gray shaded area indicates the reused time period for the scenario without overshoot.

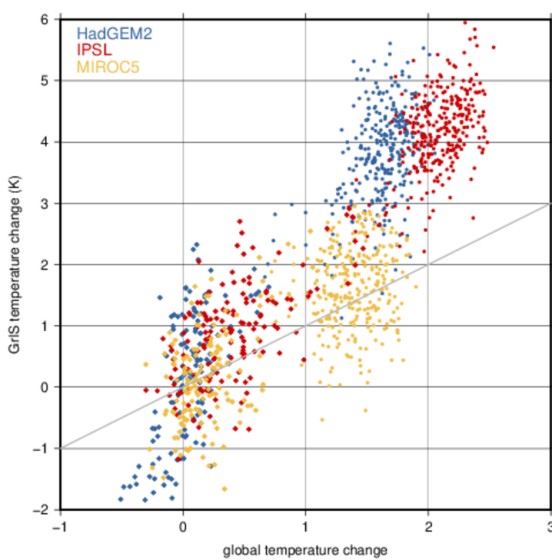

**Figure 2.** Scatter plot of annual mean near-surface air temperature change relative to pre-industrial levels over GrIS versus annual global mean near-surface air temperature change for the years 1861–2299. The gray line is the identity.

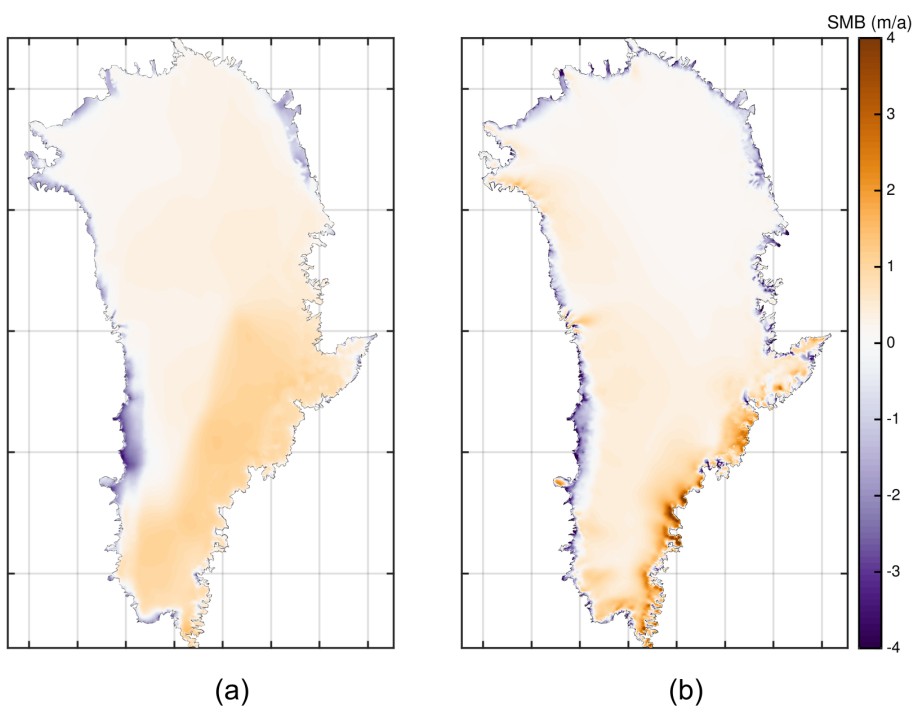

**Figure 3.** Comparison of surface mass balance fields averaged for the time period 1960–1990; (a) surface mass balance derived by forcing SEMIC with climate data from HadGEM2; (b) surface mass balance of RACMO2.3 (Noël et al., 2018).

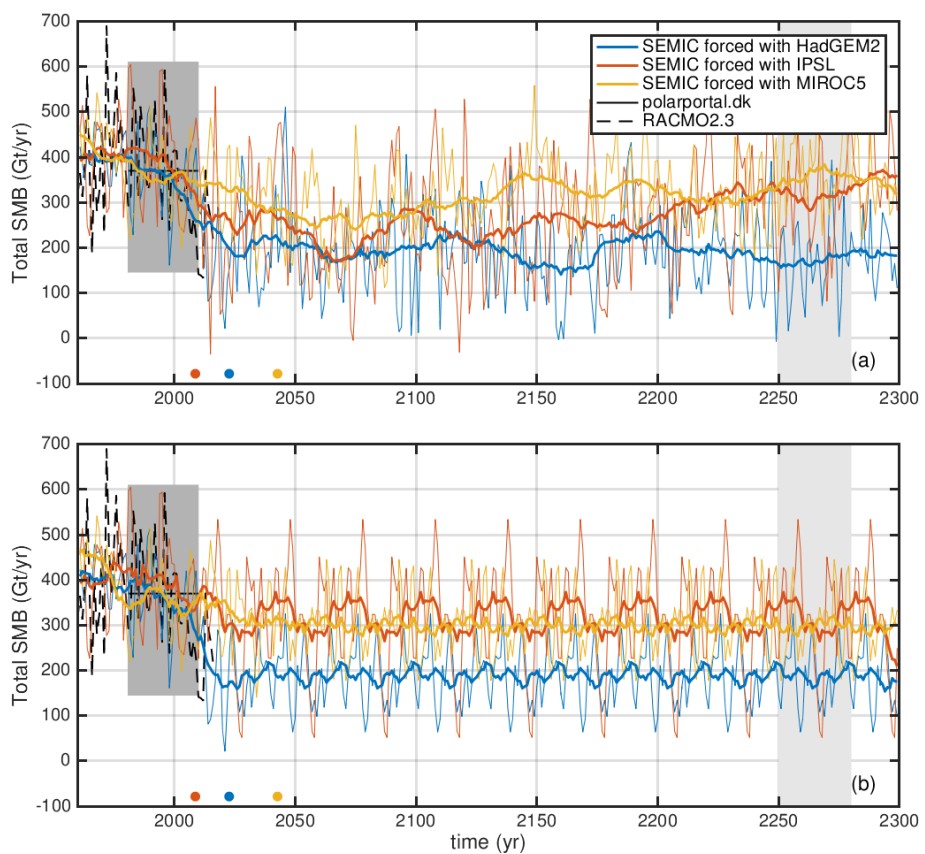

**Figure 4.** Time series of the annual mean integrated $SMB_{clim}$ ($Gt\,yr^{-1}$) according to Eq. 4 for all three SEMIC-GCMs under RCP2.6 forcing (a) and RCP2.6 forcing without overshoot (b). The solid line is a 30-year and 15-year moving mean in (a) and (b), respectively. In gray shade and black line the range and mean of SMB between 1981–2010 from Polarportal is marked (polarportal.dk). The dashed line shows the SMB time series of RACMO2.3 (Noël et al., 2018) from 1958-2016. The colored dots represent the onset years of overshooting $1.5°C$ in the global mean near-surface air temperature in a 30-year moving window relative to pre-industrial levels. The light gray shaded time period indicates the repeated SMB forcing taken from the RCP2.6 scenario for the scenario without overshoot.

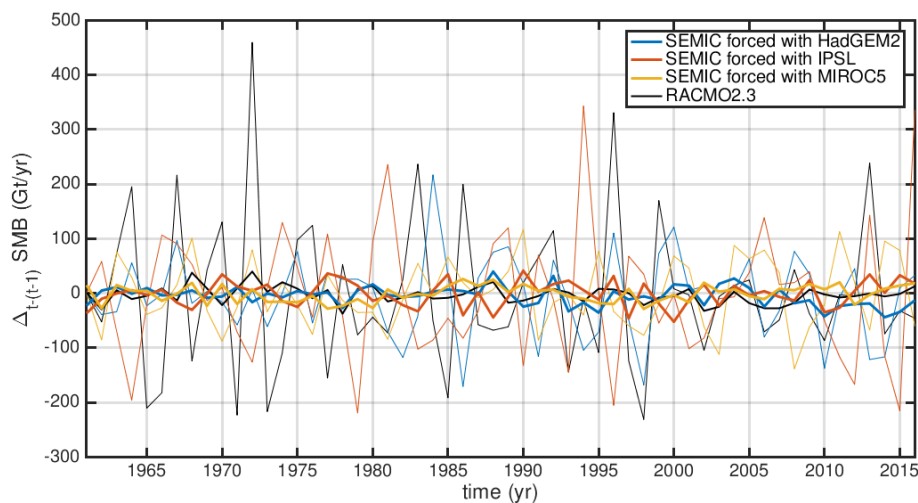

**Figure 5.** Interannual SMB variability for all SEMIC-GCMs (colored lines) and RACMO2.3 (black line) calculated from consecutive years, $\Delta\mathrm{SMB} = \mathrm{SMB}_t - \mathrm{SMB}_{t-1}$. The thick lines are a 30-year moving mean calculated from the yearly data (thin lines).

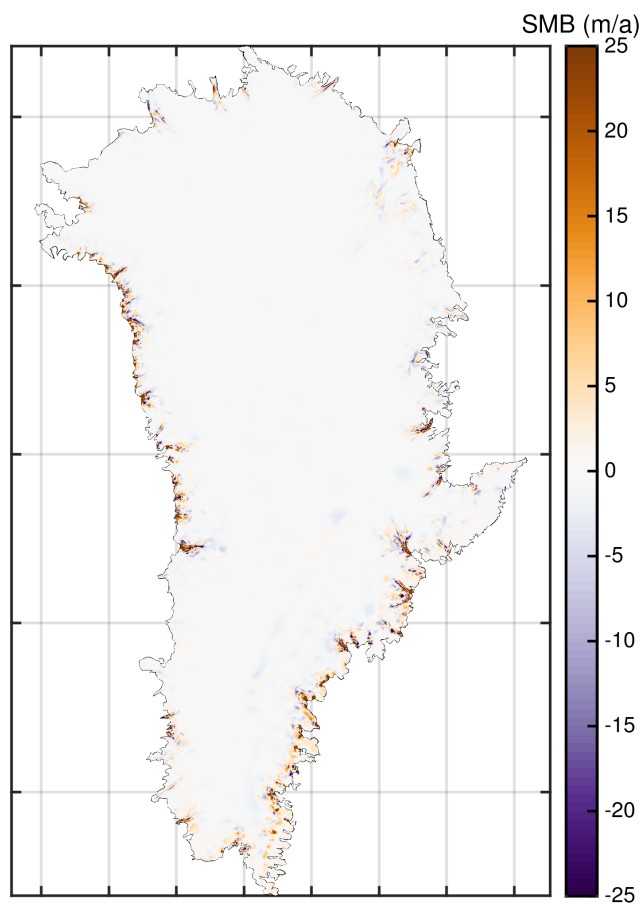

**Figure 6.** Synthetic surface mass balance $\mathrm{SMB}_{\mathrm{corr}}$ calculated from a single year unforced relaxation run (truncated at -25 and 25 $\mathrm{m\,a^{-1}}$). As the $\mathrm{SMB}_{\mathrm{corr}}$ will be subtracted in Eq. 8 positive values represent enforced thinning; negative values thickening.

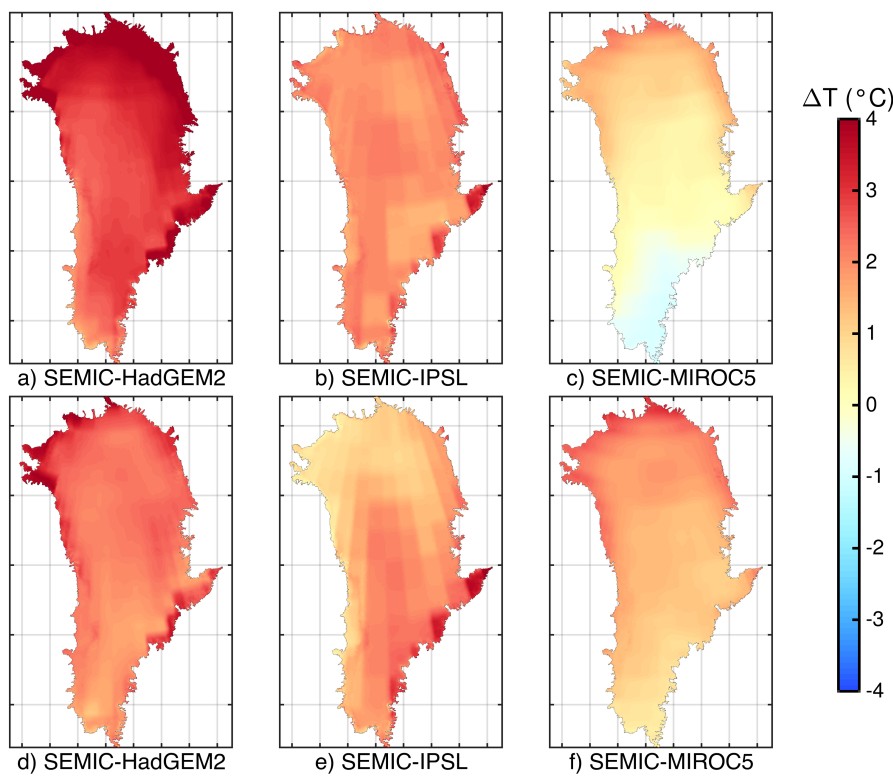

**Figure 7.** Comparison of multi-year mean surface temperature ($T_s$) differences between 2100-2000 (top row) and 2300-2000 (bottom row) for (a, d) SEMIC-HadGEM2, (b, e) SEMIC-IPSL and (c, f) SEMIC-MIROC5. The black contour line depicts the present-day ice mask.

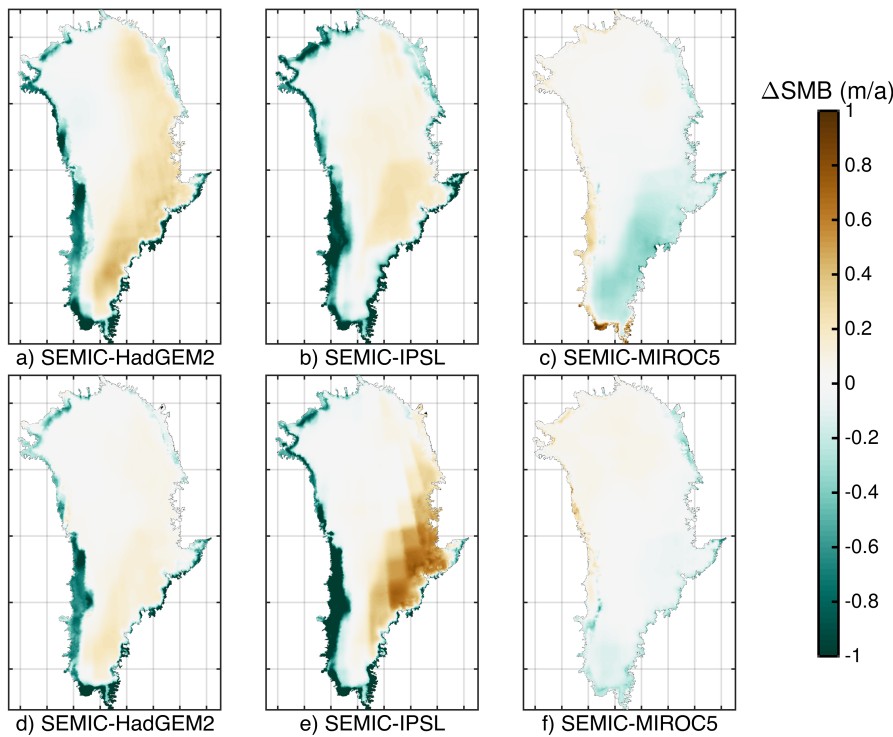

**Figure 8.** Comparison of multi-year mean surface mass balance (SMB) differences between 2100-2000 (top row) and 2300-2000 (bottom row) for (a, d) SEMIC-HadGEM2, (b, e) SEMIC-IPSL and (c, f) SEMIC-MIROC5. The black contour line depicts the present-day ice mask.

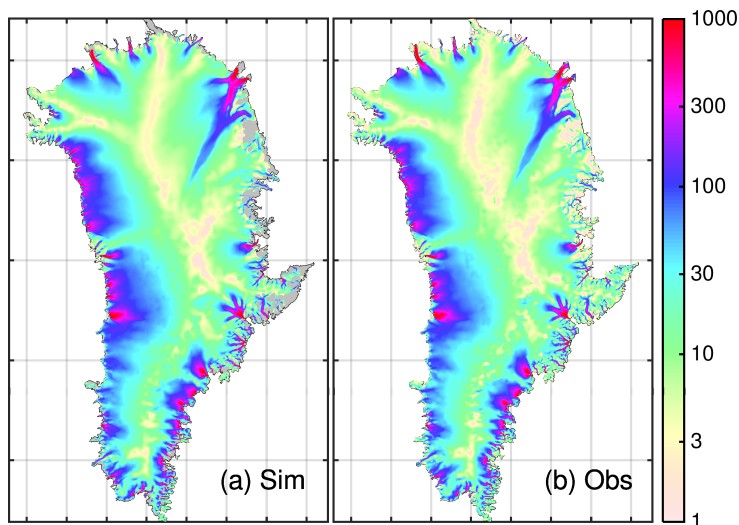

**Figure 9.** Present day velocities (year 2000) using SEMIC-HadGEM2: (a) observed velocities, (b) simulated velocities. Observed velocities: Rignot and Mouginot (2012).

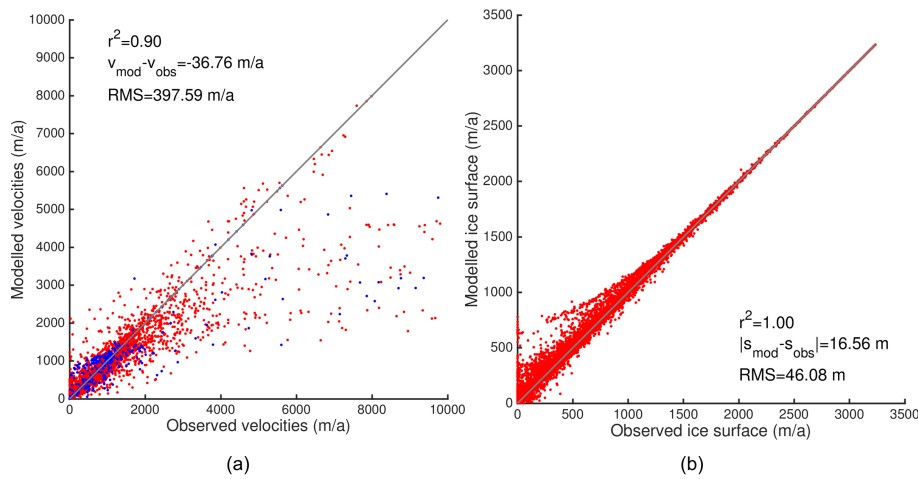

**Figure 10.** Scatter plots of the present day state (year 2000) using the SMB forcing SEMIC-HadGEM2: (a) velocities, (b) ice surface elevation. Blue and red dots in (a) represent floating and grounded points, respectively. Observed velocities: Rignot and Mouginot (2012); Observed surface elevation: Morlighem et al. (2014). The gray line depicts the identity.

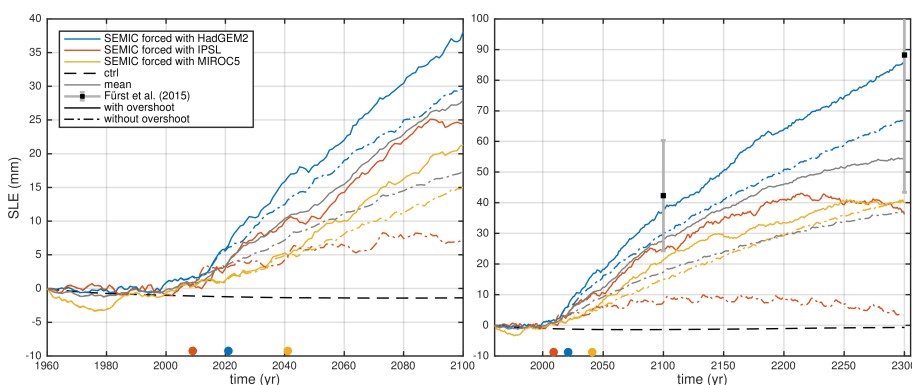

**Figure 11.** sea-level equivalent (SLE in mm) until the year 2100 (left panel) and 2300 (right panel) under RCP2.6 forcing (Solid lines) and RCP2.6 forcing without overshoot (dotted-dashed). Additionally the control run (black dashed line) and the model mean and rms deviation from Fürst et al. (2015, Table B1) are shown. The colored dots represent the onset years of overshooting 1.5°C in the global mean near-surface air temperature in a 30-year moving window relative to pre-industrial levels.

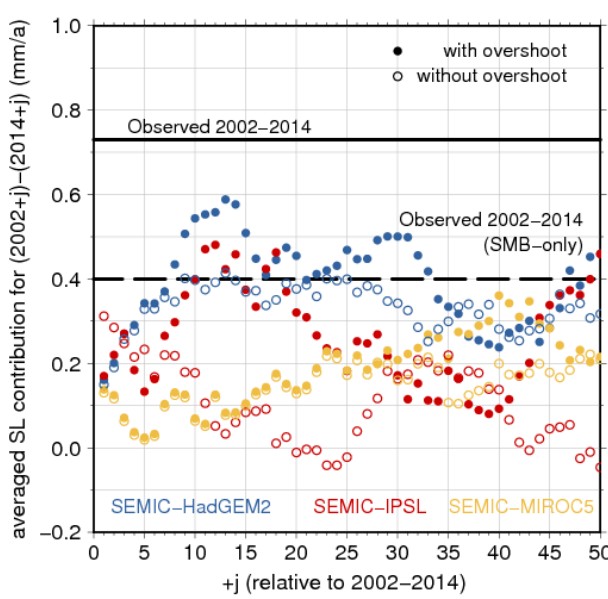

**Figure 12.** Lag (j) of projected sea-level rise per year under RCP2.6 forcing (colored dots) and the modified RCP2.6 forcing without overshoot (colored circles) as mean for a time period similar to the observational period (2002–2014). The solid black line indicates the observed value of $0.73 \, \mathrm{mm \, a^{-1}}$ by Rietbroek et al. (2016) and the dashed line the observed value of $0.40 \, \mathrm{mm \, a^{-1}}$ calculated from RACMO2.3 for the period 2002-2014.

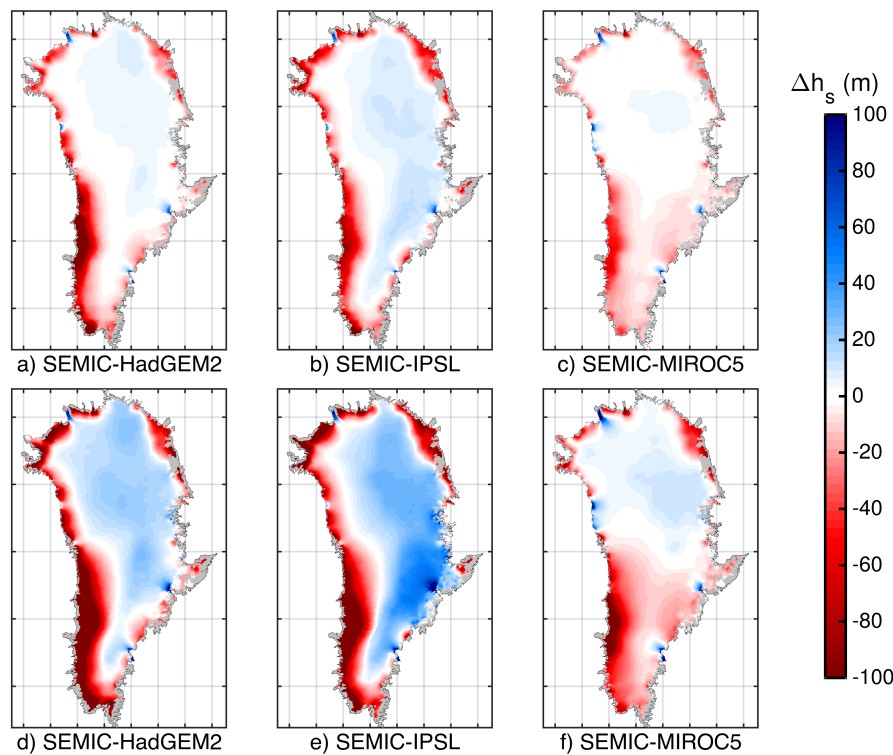

**Figure 13.** Comparison of multi-year mean surface elevation ($h_s$) differences under RCP2.6 forcing between 2100-2000 (top row) and 2300-2000 (bottom row) for (a, d) SEMIC-HadGEM2, (b, e) SEMIC-IPSL and (c, f) SEMIC-MIROC5. The black contour line depicts the present-day ice mask. Positive values represent glacier thinning; negative values thickening. The data is clipped at an ice thickness of 10 m (gray shaded area).

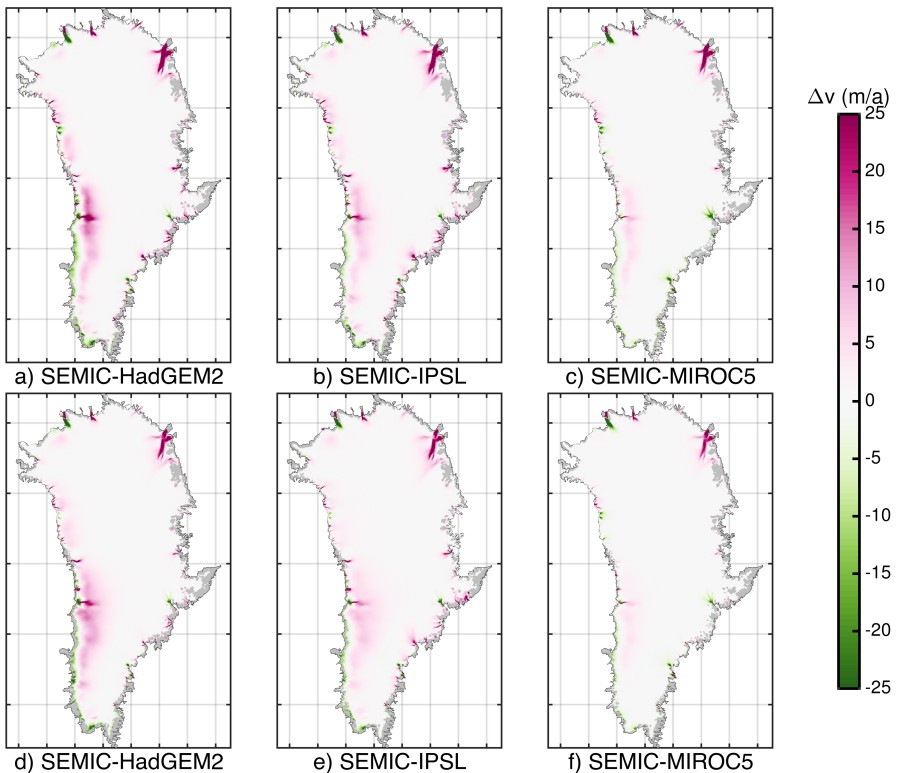

**Figure 14.** Comparison of multi-year mean surface velocity ($v$) differences under RCP2.6 forcing between 2100-2000 (top row) and 2300-2000 (bottom row) for (a, d) SEMIC-HadGEM2, (b, e) SEMIC-IPSL and (c, f) SEMIC-MIROC5. The black contour line depicts the present-day ice mask. Positive values represent glacier acceleration; negative values deceleration. The data is clipped at an ice thickness of 10 m (gray shaded area).