# Peer review of "The effect of overshooting 1.5°C global warming on the mass loss of the Greenland Ice Sheet"

_Earth System Dynamics, 2017_

## Referee Comment (RC1) · Anonymous Referee #1 · 19 Dec 2017

— Summary —

The response of the Greenland ice sheet (GrIS) to a RCP2.6 global warming scenario is studied with an ice sheet model forced by a combination of climate models. The output from existing Global Coupled Climate Model (GCM) simulations is further processed with a surface energy balance model of intermediate complexity to generate surface mass balance and temperature forcing for the ice sheet model. While a feasible two-way coupling strategy between GCMs and ice sheet models remains unavailable, this study applies anomaly forcing and a number of corrections to estimate the future sea-level contribution from the GrIS.

[Figure]

The full potential of the high-resolution, higher-order ice sheet model is not realised due to a lack of important forcing mechanisms (ocean) and a rather crude climate forcing. This leaves the application of the surface energy balance model of intermediate complexity as the main novelty compared to state of the art projections. Nevertheless, this component has not been treated with sufficient detail and its output requires more analysis and a better comparison with observations. The description of the experimental setup and processing of the forcing data is not always easy to follow and also needs more precision. I therefore suggest major revisions along the lines of my comments given below.

— General comments —

The SMB forcing is clearly the most important ingredient for this type of projection, in particular since the study does not consider any oceanic forcing. Consequently, more effort has to go into understanding and discussing the SMB product resulting from a chain of different models and processes. What is missing entirely is a (spatially resolved) validation of the used SMB forcing compared to observations and other modelling results.

The modelling approach of using the intermediate complexity model SEMIC to calculate SMB based on GCM input for projections of the GrIS sea-level contribution is one of the new and interesting aspects of this study and should receive much more attention. SEMIC is treated in the description and analysis practically as a black-box element, but should instead have a much more prominent place. The key question this study should be in the position to answer is if and why SEMIC is an improvement to, or similarly suited as other methods that are used to produce SMB forcing based on GCM output. The current alternatives include e.g. regional climate models (which are hardly mentioned in the manuscript) and models based on the positive-degree-day method.

The authors rely on the parameter settings of the SEMIC model, which have been optimised for a different climate model input (Krapp et al., 2017). The Krapp et al.
study shows that the SEMIC model can well approximate the MAR SMB results given MAR climate input. It must however be expected that the parameters that were chosen for a completely different climate input (different model, RCM vs GCM) are not optimal. Unless evidence can be provided that the applied parameters are indeed suited for the GCM forcing used in the present study, the model parameters should be optimised. Discussion on differences to other results (e.g. as done compared to Fürst et al., 2015) hinges on the implied sensitivity of the SMB model, which is currently not possible to be judged.

Modelling decisions, in particular those concerning the chain of processing used to arrive at the SMB and temperature forcing have to be better explained and motivated. In the current manuscript, some of the modelling choices appear arbitrary and it is not clear if they are optimal, possible to improve or just used in absence of better options.

The organisation of the material in the manuscript is not optimal and could profit from a reorganisation. To name just a few examples, some aspects belonging to model setup and initialisation appear too late in the text, while some results first appear in the conclusions after they have already been discussed. The ice sheet model is introduced first (2.1), while it is the much less important component for the projection compared to the SMB forcing. See also specific comments below.

There may be a problem with the thermodynamic model used to spin up the temperature as presented in Table 2. I suggest to thoroughly check and verify that aspect of the modelling.

The manuscript is so far rather short and could easily accommodate additional material that would be required to respond to the issues raised above and below.

— Specific comments —

p1 l6 Not clear why a threshold of 1.5C is relevant when calculated regionally for Greenland. To start with, the global threshold of 1.5 is a political target and is not directly

related to a real threshold in the climate system. Locally, a 1.5 degree warming has no specific meaning at all. Over which area is the Greenland wide average calculated?

p1 l8 How is plausibility of the future forcing assessed? This has to be made clearer and the wording should be changed accordingly.

p1 l14 It is not well documented what the reason for the loss of floating ice tongues really is. In the absence of ocean forcing this should be explained by interaction with the SMB. Or are part these changes related to the unforced response of the ice sheet model?

p1 l14 What values? Greenland sea-level contribution? Elevation changes? Clarify

p1 l14 A lower bound of what? The actual future sea-level contribution of Greenland? The contribution under forcing scenario RCP2.6? I think you cannot make a meaningful statement about a lower bound based on the results of this study. There is a combination of missing important processes (ocean forcing) and uncertainties about the climate forcing (intrinsic and not properly studied) that make a quantitive statement very hard to justify.

p1 l19 Repeated "past decade". Compare also "past decades" in l21. More precision needed.

p1 l22 Remove "Obviously"

p2 l2 To asses *all* "the impacts of global warming of 1.5C ..." is a huge aim. Be more specific about the aims of this study in particular.

p2 l3 RCPs were not designed for a specific warming level. Reformulate.

p2 l5 "are not passing the limit". Which limit, be more precise.

p2 l6 Remove "potential". If the effect is return to below the threshold, it is an actual overshoot.

[Figure]

p2 l9 Repeated "response"

p2 l9 Maybe "GCM" is better than "atmospheric model" here.

P2 l10 Maybe "surface mass balance changes".

P2 l12 Replace "uncoupled" by "one-way coupled".

p2 l13 The causality in this sentence is not clear. What does higher-order physics have to do with corrections of atmospheric forcing?

p2 l16 "the low computation cost"

p2 l17 Why is high resolution a requirement for higher-order physics? Also, for this study, representing the SMB forcing accurately should be the most important aspect where computational resources should be directed to.

p2 l19 "anomalies *of*"

p2 l21 More precision needed to replace "obtain these anomalies from the GCM"

p2 l24 Consider describing the ice sheet model later since it is the least important component in this study.

p2 l25 I suggest a less technical description here, e.g. "Ice flow and thermodynamic evolution of the GrIS are approximated"

p2 l28 It is not the elements themselves (as in finite elements) that have these characteristics (SIA to FS). Reformulate. Which approximation is finally used?

p3 l1 The reader does not necessarily know what "the balance equations" refers to.

p3 l5 Better to describe how basal melt rates are calculated before saying that they are held constant during the experiment.

p3 l7 "Under grounded ice"

p3 l7 Melting is not *due to* frictional heating. Frictional heating and geothermal heat

flux warm the ice that may eventually melt. More precision needed.

p3 23 "shearing"

p3 26 Remove "fields".

p4 l1 Replace "or" by "and".

p4 l1 "All methods are suitable ...". I don't think this represents the conclusions of the study very well. There are clearly methods that are more suitable than others and a combination between different methods may be needed, is how I would put it.

p4 l5 What exactly is initialized over 50 years? Is the geometry relaxed? What constant temperature is used? Be more precise in your description. The aim should be to make the model setup reproducible for other modellers.

p4 l7 Why is the spinup done to 1960, and the reference period 1960-1990. Motivation needed.

p4 l7 "basal-friction inversion" requires some additional description and references to place what is meant here in the context of state of the art techniques. What is inverted for and by optimisation of what precisely?

p4 l9 "mesh refinements are made at certain points during the initialization ..."

p4 l10 Explain better the sequence of runs. Is the forcing over 125 kyr repeated several times? The number of years add up to 290 kyr, but the forcing is supposedly only for 125 kyr.

p4 l20 What precisely is taken, thickness and bedrock data? Removes "bed from". Add "data set" after "BedMachine Greenland"

p4 l21 This belongs to the description of basal-friction inversion that should be added in the section before.

p4 l23 Add "spatially constant" before "surface temperature anomaly". Describe better

what "based on" means. Supposedly the present day RACMO temperature is offset by a spatially constant temperature anomaly?

p5 l4-7 Reformulate this sentence, too long.

p5 l10 Motivate the choice of models. Why these three GCMs?

p5 l14 Specify the reference period against which the change is calculated.

p5 l19 Could give a more specific reference here, i.e. a specific IPCC chapter.

p6 l2 Why would polar amplification only have consequences in extreme years? Or does it have an impact on the amount of extreme years? Clarify.

p6 l2 Add reference to figure 2 at end of sentence.

p6 l3 Add "amplification" after similar.

p6 l4 Polar amplification is not the same as Greenland amplification. Consider and discuss the difference and similarities if any.

p6 l7 "A striking feature" in which model?

p6 l9 "lower bound" and "upper bound" is the wrong wording for this case. Use "the highest" and "the lowest forcing" or similar.

p6 l11 "might be different across the GrIS". Why "might", you have the data to check that and make an informed statement.

p6 l13 How does a model "best" represent overshooting. Either temperature overshoots or it doesn't. Reformulate.

p6 l15 Specify what you mean by "ice sheet specific quantities".

p6 l15 It would be useful to describe the SEMIC model in coarse lines here, since it is an important ingredient to the simulations. In my opinion it represents one of the interesting new aspects in the presented simulations. Based on this description you

should judge the advantages and shortcomings of this approach and compare it to other used methods like positive-degree-day models, RCMs and other intermediate complexity models (e.g. REMBO, Robinson et al., 2010).

p6 l18 As mentioned before, SEMIC has been tuned to reproduce MAR SMB given MAR climate forcing. It cannot be expected that the model tuning translates to another model like the GCMs used here. The ultimate test is if the SMB produced for the recent past compares well against observations. This should be shown for the three GCM models and eventually it requires returning of SEMIC for that purpose.

p6 l18 Not clear what the shortcomings of the Krapp method to treat albedo were and neither how this has been improved for the present study. This requires some additional description. Extending on the last comment, changes to the albedo scheme likely also have an impact on the SMB and would lead to different tuning even for the same climate model input.

p6 l24 Motivate why this two-step procedure is necessary.

p6 l28 Add "(.)" after "quantities".

p6 l30 In my understanding $hs^{ISSM-pd}$ should be replaced by $hs^{SEMIC-pd}$. Or are they both considered the same? Please clarify.

p6 l31 What (and when) exactly is the present-day surface elevation referred to here?

p7 l4 The following three paragraphs are only remotely related to the atmospheric forcing and would fit much better with 2.2 about the initial state of the ice sheet model.

p7 l5 This is confusing. Before ISSM is run forward in time, wouldn't it have exactly the geometry that you have prescribed? A good match with the observed geometry is therefore not a result. Reformulate?

p7 l7 Remove "perfect" before equilibrium.

p7 l10 Not clear why the models have to be "run on the same ice sheet mask". Clarify.

p7 l15 Replace "ice sheet models" by "initial states".

p7 l16 Shouldn't the imbalance be subtracted to counteract it? See also equation (3), which should have a minus sign before SMB_corr.

P7 l17 The SMB correction method has been used by other modellers before (nevertheless, it is not unproblematic), which calls for adding some references (e.g. Price et al. 2011, Goelzer et al., 2013). The magnitude of the required correction should be quantified (see references above for comparison) and the shortcomings of the method should be discussed.

P7 l17 It is not clear to me why SMB_corr is time dependent here. In my understanding, the most effective method should be to subtract the imbalance diagnosed for t=1 for each year of the forward experiments (unless an iterative procedure is used). What SMB_corr is used after the end of the relaxation run from 2060 onwards? Please explain this better.

p8 l3 "GCM" does not appear in the formula.

p8 l4 I thought RCP2.6 was only defined until 2100. Describe how it has been prolonged if that is what has been done here.

p8 l5 Maybe "albeit without a correction term"?

p8 l9 What does "bias corrected onto the [..] grid" mean exactly?

p8 l14 "respectively".

p9 l16 In my understanding h_fix should be the modelled present-day surface elevation, not the observed. This would result in corrections for the actually occurring elevation changes. Or are they (modelled and observed) identical?

p9 l25 These gradients were found as best fit to SMB simulated by a specific RCM (MAR) at different elevations. Applying these in your setup may be better than nothing, but for a consistent picture, these should ideally be recalculated based on your own

model setup (SEMIC). Maybe, if you can run SEMIC at different elevation, you could get a feeling for the implied differences. At the very least this inconsistency should be recognised and discussed as a shortcoming.

p9 l6 replace "reveals" by "shows" or "exhibits"

p9 l12 What criteria are used to judge plausibility of the warming patterns?

p9 l15 Same problem here. What criteria are used to judge implausibility of the warming patterns?

p9 l16 Add "as Figure 3" after "in a similar fashion".

p9 l16 Remove "as" before "as" or "as" after "as".

p9 l22 Reformulate "extreme pattern".

p9 l33 Validation of the SMB for the present day has to come much earlier to give confidence in SEMIC and should include analysis of the 2D pattern, not only total numbers.

p10 l2 All of this suggests that the confidence in the derived SMB forcing (and consequently the resulting SL numbers) is rather low, something that should be discussed in the end of the paper. However, ultimately you are using anomalies with respect to 1960-1990, so maybe that looks better. To be shown.

p10 l11 Is it important which model is used? If not, make that clear.

p10 l15 These results are difficult to see in Figure 6. It could help to plot velocity differences or ratios instead. Zooming in on some important regions could also give the interpretations more substance.

p10 l19 This paragraph should start with a motivation before going into technicalities on how things are calculated.

p10 l24 It seems like a strange choice to not correct the reported SL changes for the

model drift. I interpret all the corrections that go into the method as an attempt to produce a steady state at 1960. Or are you suggesting that the model drift should represent some natural background evolution? In my understanding the (negative) SL response in an unforced forward experiment is purely an artefact of the initialisation method and should be corrected. Another motivation would be to be transparent about the remaining model drift, which I could appreciate. However, in this case the results of a full control experiment should be presented alongside with the SL numbers of the forced experiments so that the actual magnitude of the projection can be easily judged by the reader.

p10 l25 As mentioned in the general comments, I am not convinced that the timing when Greenland mean temperature changes cross 1.5 degree is a very meaningful diagnostic, in the light of spatially divergent warming trajectories. What interpretation are you hoping to derive from this analysis?

p11 l4 "This is potentially an effect of ice dynamics"? You are running an ice sheet model, which should put you in the place to make an informed statement about what is going on here.

p11 l9 Reformulate "false trend".

p11 l18 What are these "errors in vertical ice velocities"? If this is a shortcoming of your ice sheet model, that should be discussed at some place in the model description. Does the same problem occur in the unforced control experiment? Again, being in full control of the ice sheet model in use here, you should be able to diagnose exactly what the problem is.

p11 l27 Why is this section called "Acceleration" when some of the glaciers see deceleration? I suggest rewording to "Dynamic response" or similar.

p11 l32 I am wondering in how far a detailed analysis of individual glaciers is justified given that an important aspect of the forcing in form of interaction with the ocean and

sub-glacial hydrology is missing. The comparison suggests that we could hope to get the behaviour of individual glaciers in line with observations, which I consider very unlikely given the steady-state initialisation, coarse GCM-based forcing and lack of important forcing mechanisms.

p12 l2 You could speculate that you could maybe reproduce observed acceleration of Jacobshavn Isbrae if calving rates are forced like in Bondzio et al (2017). If this is really the case in your model is not clear until you have tried it. Reformulate.

p12 l7 What is generally the magnitude and pattern of the SMB correction, average, largest magnitude, overall positive or negative? Where is it particularly prominent? What does that mean for ice dynamics and SMB, which fail to generate or export enough ice from a given region?

p12 l7 Replace "undermining" by "underlining"

p12 l10 What does "geometric settings at their base" refer to? Clarify

p12 l10 Why does alternation between acceleration and deceleration mean the model is able to "resolve glacier valleys well"? What does it mean to resolve glacier valleys well? The geometry, the velocity structure within the valleys?

p12 l14 Sea-level contribution is in mm not mm a-1

p12 l31 These numbers should be given before, when the results are being discussed, and as mentioned earlier, together with the model drift of an unforced control experiment.

p13 l2 This paper requires a dedicated discussion section before the conclusions that serves to discuss the advantages and shortcomings of the models and processing steps needed to arrive at the final numbers.

p13 l4 "switching between spin-up and RCP forcings" A correctly applied anomaly method should not lead to any additional model drift, other than the imbalance resulting from imperfection of the data assimilation process. Possibly the SMB implied during initialisation differs from the one used further on? Often modellers use a (short) relaxation run as part of the initialisation to avoid too large model drift in the forward experiments, possibly combined with a correction method as applied here. At any rate, the uncorrected model drift of as much as 50 % of the signal by 2100 (MIROC) and the corrected model drift of still 30 % of the signal seems pretty large given the low magnitude RCP2.6 forcing applied here. This should be discussed in the paper at some point.

Table 1 Not clear which actual years are covered by these spin-up runs. Clarify.

Table 2 - What does it mean when a temperature of 0.00 is indicated as modelling results? The -2.4 at NGRIP means that the temperature is at the pressure melting point (PMP). Is that the case for the simulated temperatures for p-cl,Gr and pd-cl,Gr? - Basal temperatures of ∼-20 seem to be extremely low compared to the observed ice core temperatures (nowhere below -14) and are at odds with my own experience in thermodynamic modelling of the GrIS. The results should raise some doubts about the correctness of the applied thermodynamic model. - Typically, one would expect the pd spinup to result in generally warmer basal temperatures throughout, because of the lack of glacial signatures in the evolution. This is not confirmed in some cases. Why is that? - Could add the NEEEM ice core to the list of constraints

Figure 1 Add what area is used to calculate GrIS warming. All land area, observed ice sheet mask? b) Include GrIS in y-label.

Figure 2 Caption: "The grey line depicts the identity" Also describe here which range of years are plotted and from what product (grid). Add what area is used to calculate GrIS warming.

Figure 3 Colour bar labels are not well readable at this size. Could remove identical colour bars per row of figures and have one big one.

[Figure]

Figure 4 Colour bar labels are not well readable at this size. Could remove identical colour bars per row of figures and have one big one.

Figure 5 The forcing that the ice sheet model actually sees and that goes into the SL projections is based on anomalies of the SMB with respect to 1960-1990. How does figure 5 look like and how does the constructed SMB compare to observations when this anomaly calculation is applied? Caption: Is there a paper reference available for the SMB observation product?

Figure 6 Figure colour bar labels are not well readable at this size. Could remove one of the identical colour bars per row of figures. Should add contour lines in panel c and d. Caption: (a) simulated horizontal velocity magnitude, (b) observed horizontal velocity magnitude (Rignot and Mouginot, 2012), ...

Figure 7 Figure labels are not well readable at this size. Labels should be increased to be readable in the final two-column layout. Caption: Add what area is used to calculate GrIS warming. You should note here that the relaxation run differs in setup from the other experiments.

— References —

Price, S. F., Payne, A. J., Howat, I. M., and Smith, B. E.: Committed sea-level rise for the next century from Greenland ice sheet dynamics during the past decade, Proc. Nat. Acad. Sci. U.S.A., 108, 8978-8983, doi:10.1073/pnas.1017313108, 2011.

Robinson, A., Calov, R., and Ganopolski, A.: An efficient regional energy-moisture balance model for simulation of the Greenland Ice Sheet response to climate change, The Cryosphere, 4, 129-144, 2010.
* * *

---

## Referee Comment (RC2) · C. Schannwell (Referee) · 20 Feb 2018

Review of Rückamp et al. "The effect of overshooting 1.5C global warming on the mass loss of the Greenland Ice Sheet."

**General comments:**

This manuscript presents future volume evolution scenarios of the Greenland Ice Sheet under three different surface mass balance forcings. Atmospheric forcing is provided by three global climate models and the surface mass balance is computed with a relatively simple surface energy balance model. The ice-sheet model employed, is the state-of-the art ISSM model with higher order ice physics. The sea-level rise projections from surface mass balance perturbation alone are between 46-71 mm by 2100 and 114-189 mm by 2300.

The topic of the manuscript is of interest to ice-sheet modellers as well as the wider cryospheric community. The overall structure of the paper is logical but some sections would benefit from a tidy-up and the language is hard to follow in some places. While the results are certainly not groundbreaking and omit any contributions from ice dynamics, I think the manuscript presents enough novelty and hence merits publication subject to consideration of my comments listed below.

**Specific comments**

The study's strong point from an ice-sheet modelling perspective is the model initialisation which combines the two commonly employed spin-up and data assimilation techniques. The main focus is, however, on the surface mass balance forcing with the SEMIC model. In the light of this and the importance of the surface mass balance forcing, for someone that is not familiar with the SEMIC model, I am missing a succinct description of the model fundamentals and the configuration used in this manuscript. Furthermore, the entire manuscript would benefit from some reordering and substantial improvements to certain sections and improvements in readability of some figures (detailed below). My main concern is with the calculation of the surface mass balance anomaly for the projections. Please find below my main concerns, followed by specific comments.

**Main concerns:**

1. My main concern is the calculation of the surface mass balance anomalies. First of all, I understand that you account for the model drift by adding a synthetic SMB correction term ($SMB_{corr}$ in Equation 3). But what dh/dt is applied – an average of your unforced relaxation run from 1960-2060 or the last or first time step of this relaxation simulation? How can this term be time-varying in your projections? On page 9 line 20 this time-varying $SMB_{corr}$ term is used as an explanation for spatial differences in the SMB pattern. Maybe I missed it, but it would help if you clarified this.

2. The more critical point is how you compute your $\Delta SMB$ in Equation 3. The way I understand it and please correct me if I am wrong, Equation 3 states that SMB_RACMO plus your correction for the model drift should give you an SMB that keeps your ice sheet close to steady state (or at least present geometry). The applied perturbations are however calculated with respect to the SEMIC model baseline. If

you use your RACMO_SMB to keep your ice sheet in steady state, you should also calculate your anomalies with respect to your SMB_RACMO field. If not, your perturbations to the surface mass balance appear a bit arbitrary. Would it not be more consistent to use the SEMIC output? The argument that your model drift gets larger is rather weak, considering that you would just get a larger $SMB_{corr}$ term from the unforced relaxation simulation.

3. I think the section "Input data" should be removed as this mostly repeats earlier statements (e.g. Greve 2005 dataset). The basal drag inversion should be moved to the "Initial state" section as this is where it is most appropriate. I would introduce a section "Results" which would start with the subheading "Forcing fields" and continue with "Present day elevation and velocities". The heading "Projections" followed by "Present day …" was confusing. I would suggest to add "projections" where appropriate e.g. Mass loss projections, Speed up projections etc.

4. Please provide a more complete description of the SEMIC model than the few lines provided on P6 L15-22. You also claim to have improved the albedo parameterisation, but to me it is not clear how or to what extent. Please expand on this.

5. I am certainly not an expert on ice temperature, but to me the following questions came up when looking at Table 2. Are there no temperatures from observations for EastGRIP? Why are there such large differences in basal temperatures between the Greve (2005) and Shapiro and Ritzwoller (2004) maps at the selected locations? Does this mean that temperature in these regions is dominated by the geothermal heat flux and that this heat flux is that different at these locations? Why do the simulated temperatures do not agree with GRIP temperature observations?

**Technical corrections**
**Abstract**
L2 "…sea-level change under different atmospheric forcing scenarios from …"
L11 Sentence starting with "Simulated an observed sea-level rise…" That makes no sense to me. Is it simulated or observed? I believe you are trying to say that your simulated sea-level rise for the period 2002-2014 matches sea-level rise from observations in magnitude? Please clarify.

P1L19 delete second "past decade"
P1L22 Delete "Obviously, …"
P2L1 „engaged"? Do you mean encouraged?
P2L20 "…provided by …"
P2L27 replace "." with ","
P2L27 Sentence starting with "ISSM is designed to … " Is this really important for the paper? Also while I welcome the fact that the authors kept the details of the ice-sheet model brief, I would appreciate if you could add what higher-order physics you used (Blatter-Pattyn, Stokes or SSA)? Please add to ice-flow model section. Also, can elements be either Stokes or SIA? Do you mean that for each element you can choose what force balance is solved?
P3L2 "…surface mass balance and climate forcing"
P3L19 "compensates"

P3L20-21 "…according to a sub-grid paramterization scheme,…"

P3L24 "… towards the base where vertical shearing becomes more important."

P4L4 Delete sentence starting with "Furthermore, the themo-mechanically …" I think it is obvious that if you simulate ice temperatures that your simulations are sensitive to temperatures.

P4 L5-13 and Table 1 I do not completely understand when you start your mesh refinements? The way I understand your initialisation method is that you run your temperature spin-up with mesh sequence 1, then you do an inversion for basal friction parameters and run your temperature spin-up again with a refined mesh before you do another inversion on the refined mesh? Please describe this more clearly.

P5L4-7 Please reformulate this sentence. It is too long. Also please delete "aim" as this implies that you are not sure it is going to work. Your results show that it clearly does work.

P5L10 Could you explain why the three GCMs were selected as forcing? So far this choice appears a bit random.

P5L20 This sentence is unclear. It reads like Greenland warms above 1.5°C but you are talking about ∆T I believe. Also, could you state more clearly that you are comparing it to the global temperature increase in the GCMs..

P6L3 Sentence starting with "While HadGEM2 … " makes no sense to me. Leading to similar factors? What factors?

P6L8 "reaches"

P6L8-9 This sentence has to come earlier as it is indeed very striking, but also expected.

P6L9 Please delete "Summarizing"

P6L17 Please delete "Due to the fact that Krapp et al. (2017) performed calibration over GrIS"

P6L28 "We follow …"

P7L4 Here and throughout "the ISSM"="ISSM"

P7L5 very well = well

P7L14-16 This statement needs a citation. Is this true for Greenland? I doubt that every data assimilation initialization leads to a 3% ice volume gain.

P8L5 By doing so = This ensures that

P8L14-15 espectively=respectively

P9L4 "leads to an increase in temperatures …"

P9L5 "exceed 2°C of warming"

P9L6 and P9L8 Be more specific. By how much? Numbers please!

P9L13-15 Please explain this. Why is this the most plausible? It is not apparent to me.

P9L16 delete first "as"

P9L19 here and throughout vallies=valleys

P9L20-21 See main comment above. How can this be time-varying?

P9L27 "The magnitude of ∆SMB is far less in the period 2300-2000…"

P9L31 which pattern? Spatial or temporal or both?

P11L14 Again why is this the most plausible pattern? Please elaborate.

P11L33 "… experience acceleration across all simulations."

P12L8 levelled out = balanced

P12L17 "…ice sheet loses contact with the ocean."

P12L17 resolution = grid resolution

P12L28 "considerably large". What does this mean? Be more specific!

**Figures:**

Figure 1: Can you make the line for 1.5°C bold to aid visibility when the models pass this threshold?
Figure 3: Question mark before "C" symbol in Figure. Colour bar is too small. As it is the same magnitude for all panels one big colour bar should suffice.
Figure 4: See comments for Figure 3
Figure 6: Again, use one colour bar per panel. Also, please have colour bar labels on the same side of the colour bar and avoid overlap of axes labels with main Figure. Please align top and bottom panels properly.
Figure 7: Again bigger axes labels and legends.
Figures 8 and 10: See comments for Figure 3

Sincerely, Clemens Schannwell

---

## Author Comment (AC1) · 20 Mar 2018

Dear Editor, dear Reviewers,

the authors wish to thank both reviewers for the detailed review containing many helpful remarks and constructive criticism! We do appreciate very much the time spent on getting down to the study. Before answering the points raised by both reviewers, we want to inform that we have detected an application error when calculating the SMB with SEMIC. This has a significant effect on the results, for instance the SLE contribution (Figure 1).

Figure 1: Sea level equivalent until the year 2100 (left panel) and 2300 (right panel) for all GCMs. Additionally, the control run and the mean of all GCMs are shown. The right panel shows additionally the mean and standard deviation at the year 2100 and 2300 by Fürst et al. (2015).

Beside the updated results (which are now fit better to observations and previous studies) and the corresponding changes to the text, we have performed the following major changes - that are all also documented below in detail:

- As both reviewers suggested re-structuring the manuscript, the material in the revised manuscript is presented as follows:
  - 1. Introduction
  - 2. Model description
    - 2.1 SEMIC
      - 2.2 Ice flow model
  - 3. Results
    - 3.1 Forcing fields
    - 3.2 Present day state
    - 3.3 Projections
  - 4. Discussion
  - 5. Conclusion
- We have improved the control run (see Figure 1).
- SEMIC and the calculation of the SMB are explained in more detail and became more prominent in the manuscript.
- Although the point was not raised by the reviewer, we will focus more on "The effect of overshooting 1.5°C" as mentioned in the title

Technicalities: below we answer each point raised by the reviewers and mark our answer in blue color. Point raised by both reviewers are answered at one location and referenced at the second one. 'Done.' denotes that this point would be solved in the revised version of the manuscript. This could be that it will be either done directly, or that due to other changes the point does not arise any more, or that the point has been answered at another place in this text already.

**Reviewer #1**

- Summary -

The response of the Greenland ice sheet (GrIS) to a RCP2.6 global warming scenario is studied with an ice sheet model forced by a combination of climate models. The output from existing Global Coupled Climate Model (GCM) simulations is further processed with a surface energy balance model of intermediate complexity to generate surface mass balance and temperature forcing for the ice sheet model. While a feasible two-way coupling strategy between GCMs and ice sheet models remains unavailable, this study applies anomaly forcing and a number of corrections to estimate the future sea-level contribution from the GrIS. The full potential of the high-resolution, higher-order ice sheet model is not realised due to a lack of important forcing mechanisms (ocean) and a rather crude climate forcing. This leaves the application of the surface energy balance model of intermediate complexity as the main novelty compared to state of the art projections. Nevertheless, this component has not been treated with sufficient detail and its output requires more analysis and a better comparison with observations. The description of the experimental setup and processing of the forcing data is not always easy to follow and also needs more precision. I therefore suggest major revisions along the lines of my comments given below.

- General comments -

(1000 1000

The SMB forcing is clearly the most important ingredient for this type of projection, in particular since the study does not consider any oceanic forcing. Consequently, more effort has to go into understanding and discussing the SMB product resulting from a chain of different models and processes. What is missing entirely is a (spatially resolved) validation of the used SMB forcing compared to observations and other modelling results.

Yes, indeed this is an important point and we followed the reviewers suggestion. With using the parameters of Krapp et al. (2017) the direct output of the SMB from SEMIC has a misfit of about ~2m/a and a correlation of ~ $r^2$ =0.5 by comparing SMB\_RACMO\_1960-1990 and SMB\_SEMIC\_1960-1990 (almost similar for all GCMs used). However, recalling Equation 3 and 4 from the manuscript,

$$SMB(x, y, t) = \overline{SMB}_{RACMO}^{(1900-1990)}(x, y) + \Delta SMB(x, y, t) + SMB_{corr}(x, y, t),$$
(3)

(4)

$$\Delta \text{SMB}(x, y, t) = \text{SMB}_{\text{SEMIC}}(x, y, t) - \overline{\text{SMB}}_{\text{SEMIC}}^{(1960-1990)}(x, y),$$

we do not use the direct output of SEMIC, but apply anomalies computed using SEMIC. The benefit of our approach is, that only the GCM trends of SMB changes are added to the RACMO SMB reference field, which represents the real SMB distribution very well. If we compare the computed SMB to RACMO (according to Eq. 3 and 4 without the synthetic SMBcorr), for instance for the HadGEM2-ES year 1990, it shows a very good agreement (Figure 2). See also answer to specific comment "p10 I2" below. In the revised manuscript we dedicate an own section to this issue.

Figure 2: (left panel) surface mass balance of RACMO2.3 (Noel et al., 2016) for the year 1990; (middle panel) surface mass balance for HadGEM2-ES for the year 1990 according to Eq. 3 and 4 in the manuscript (without SMBcorr); (right panel) scatter plot of both fields.

The modelling approach of using the intermediate complexity model SEMIC to calculate SMB based on GCM input for projections of the GrIS sea-level contribution is one of the new and interesting aspects of this study and should receive much more attention. SEMIC is treated in the description and analysis practically as a black-box element, but should instead have a much more prominent place. The key question this study should be in the position to answer is if and why SEMIC is an improvement to, or similarly suited as other methods that are used to produce SMB forcing based on GCM output. The current alternatives include e.g. regional climate models (which are hardly mentioned in the manuscript) and models based on the positive-degree-day method.

We expand the section about the SEMIC model in order to give the reader a better understanding of the model. In the new version of the manuscript we also review in the introduction section briefly the already existing alternatives used and relate the discussion section accordingly. The reason we have not included too much detail on that issue previously is, that we basically apply SEMIC and that the model in itself and all the parameter tuning is work done by Krapp et al., 2017. The advantage of using a semi-complexity model is indeed its simplicity and cost efficiency, which would allow ice sheet modellers to also run computation up to time scales of thousands of years (e.g. until 5000) studying long-term commitment of various emission scenarios and hence not be limited by the availability of regional climate model output. However, regional climate models having the clear benefit for representing snow and firn layers with all melt and refreezing processes by far more realistic than any semi-complexity model will ever do. For the future, we plan a study on comparing the difference in ice sheet model response to three different types of forcings, PDD, SEMIC and a regional climate model forcing.

The authors rely on the parameter settings of the SEMIC model, which have been optimised for a different climate model input (Krapp et al., 2017). The Krapp et al. study shows that the SEMIC model can well approximate the MAR SMB results given MAR climate input. It must however be expected that the parameters that were chosen for a completely different climate input (different model, RCM vs GCM) are not optimal. Unless evidence can be provided that the applied parameters are indeed suited for the GCM forcing used in the present study, the model parameters should be optimised. Discussion on differences to other results (e.g. as done compared to Fürst et al., 2015) hinges on the implied sensitivity of the SMB model, which is currently not possible to be judged.

We haven chosen the same parameters of SEMIC as Krapp et al., 2017, due to the following reason: the parameter tuning procedure performed by Krapp et al., 2017 aimed to find a parameter set which gives a best fit between SMB and skin temperature Ts of SEMIC with only a limited number of processes and simpler parameterisations than a regional climate model with full complexity would derive. As a regional climate model is typically validated against reanalysis data and observations, the best match between SMB and Ts of SEMIC and regional climate model (in that case MAR) is the best way to represent the processes and their parameters in SEMIC. We see it thus as a tuning of the parameterisation of the processes. Once the process description in SEMIC is optimised, any type of input, either GCM or reanalysis data fields, will lead to the best possible SMB and Ts fields that SEMIC can produce. Still, the GCM will lack the best atmospheric fields over the ice sheet, as it is limited in resolution compared to a regional climate model. Given experiences we made from these three GCMs used in this study, which are all have different drawbacks, which would mean to have a tuning for each of them and this tuning would then make the whole benefit of having a semi-complexity model with low costs meaningless. Furthermore, it would basically mean to compensate far too low near surface temperatures with SEMIC parameters, which would offset the whole comparison of GCM forcing. Therefore, we have chosen a different approach: we compensate for this by using the SEMIC output only as an anomaly.

Modelling decisions, in particular those concerning the chain of processing used to arrive at the SMB and temperature forcing have to be better explained and motivated. In the current manuscript, some of the modelling choices appear arbitrary and it is not clear if they are optimal, possible to improve or just used in absence of better options.

We can understand this and follow the reviewer's recommendation and try to describe the processing of SMB and Temperature product better in the revised manuscript.

The organisation of the material in the manuscript is not optimal and could profit from a reorganisation. To name just a few examples, some aspects belonging to model setup and initialisation appear too late in the text, while some results first appear in the conclusions after they have already been discussed. The ice sheet model is introduced first (2.1), while it is the much less important component for the projection compared to the SMB forcing. See also specific comments below.

We do not agree that an ice model is much less important for the projection compared to the SMB forcing. In order do estimate the SL contribution from the ice sheets an appropriate ice flow model (reso

---

## Author Comment (AC2) · 20 Mar 2018

The comment was uploaded in the form of a supplement:
https://www.earth-syst-dynam-discuss.net/esd-2017-103/esd-2017-103-AC2-supplement.pdf

---

## Referee Report (RR1)

Review of "The effect of overshooting 1.5◦C global warming on the mass loss of the Greenland Ice Sheet" by Martin Rückamp et al.

**Summary**

I am evaluating this paper for the second time after major revisions in the first round. The manuscript has changed considerably since the last iteration, has improved in response to the reviewer's comments and has seen new material being added. However, some major issues remain or have been introduced with the modifications, as is the case for the validation part and the "scenarios without overshoot". The language also needs further improvements to clean up errors and make the text clearer.

I will first respond to some of the author comments (blue), with my initial comments indented twice. General and specific comments on the new version follow below.

**Response to the discussion**

> - Yes, indeed this is an important point and we followed the reviewers suggestion. With using the parameters of Krapp et al. (2017) the direct output of the SMB from SEMIC has a misfit of about ~2m/a and a correlation of ~r2=0.5 by comparing SMB_RACMO_1960-1990 an dSMB_SEMIC_1960-1990 (almost similar for all GCMs used).

This evaluation should be extended and backed with figures to appear also in the manuscript. See general comments.

> - However, recalling Equation 3 and 4 from the manuscript, we do not use the direct output of SEMIC, but apply anomalies computed using SEMIC. The benefit of our approach is, that only the GCM trends of SMB changes are added to the RACMO SMB reference field, which represents the real SMB distribution very well. If we compare the computed SMB to RACMO (according to Eq. 3 and 4 without the synthetic SMBcorr), for instance for the HadGEM2-ES year 1990, it shows a very good agreement (Figure 2). See also answer to specific comment "p10 l2" below. In the revised manuscript we dedicate an own section to this issue.

This comparison does not really validate the SEMIC model. It only shows that at a given point (year 1990) the anomalies of SEMIC are close to zero. See general comments.

> - We expand the section about the SEMIC model in order to give the reader a better understanding of the model. In the new version of the manuscript we also review in

the introduction section briefly the already existing alternatives used and relate the discussion section accordingly. The reason we have not included too much detail on that issue previously is, that we basically apply SEMIC and that the model in itself and all the parameter tuning is work done by Krapp et al., 2017. The advantage of using a semi-complexity model is indeed its simplicity and cost efficiency, which would allow ice sheet modellers to also run computation up to time scales of thousands of years (e.g. until 5000) studying long-term commitment of various emission scenarios and hence not be limited by the availability of regional climate model output.

Multi-millennial simulations are not relevant for the current study. The question is if the chosen model is an appropriate tool for the presented type of simulation. The SEMIC model has so far not been validated for the use with GCM input data, the way you are using it for the projections. Showing a proper validation is the price you have to pay for that novelty.

- The authors rely on the parameter settings of the SEMIC model, which have been optimised for a different climate model input (Krapp et al., 2017). The Krapp et al. study shows that the SEMIC model can well approximate the MAR SMB results given MAR climate input. It must however be expected that the parameters that were chosen for a completely different climate input (different model, RCM vs GCM) are not optimal. Unless evidence can be provided that the applied parameters are indeed suited for the GCM forcing used in the present study, the model parameters should be optimised. Discussion on differences to other results (e.g. as done compared to Fürst et al., 2015) hinges on the implied sensitivity of the SMB model, which is currently not possible to be judged.

We haven chosen the same parameters of SEMIC as Krapp et al., 2017, due to the following reason: the parameter tuning procedure performed by Krapp et al., 2017 aimed to find a parameter set which gives a best fit between SMB and skin temperature Ts of SEMIC with only a limited number of processes and simpler parameterisations than a regional climate model with full complexity would derive. As a regional climate model is typically validated against reanalysis data and observations, the best match between SMB and Ts of SEMIC and regional climate model (in that case MAR) is the best way to represent the processes and their parameters in SEMIC. We see it thus as a tuning of the parameterisation of the processes. Once the process description in SEMIC is optimised, any type of input, either GCM or reanalysis data fields, will lead to the best possible SMB and Ts fields that SEMIC can produce. Still, the GCM will lack the best atmospheric fields over the ice sheet, as it is limited in resolution compared to a regional climate model. Given experiences we made from these three GCMs used in this study, which are all have different drawbacks, which would mean to have a tuning for each of them and this tuning would then make the whole benefit of having a semi-complexity model with low costs meaningless. Furthermore, it would basically mean to compensate far too

> low near surface temperatures with SEMIC parameters, which would offset the whole comparison of GCM forcing. Therefore, we have chosen a different approach: we compensate for this by using the SEMIC output only as an anomaly.

I understand the argument to avoid tuning the model for individual GCMs and I agree with the point on compensating errors in the GCM. Nevertheless, I think you will agree that if you were to tune SEMIC for another RCM (say RACMO or HIRHAM), you would end up with different parameters. I believe it is important to recognise that, even if you chose to do nothing about it and use the MAR based parameters.

The situation here is worse, because using forcing from a GCM implies different characteristics, like smoother gradients and less resolved geometry compared to the RCM. It is possible that these characteristic differences between RCM and GCM (not individual model bias) have an important impact on the modelled SMB. I believe it is your responsibility to show that the parameters that you are using are indeed appropriate for the given purpose. You should show how the absolute SMB looks like for the different GCMs and compare that to reconstructions and/or state-of-the-art RCM results.

I agree that using the anomaly method is a good choice, as it circumvents *some* of the biases in the absolute SMB products you are producing with SEMIC. Nevertheless, the reference SMB has an important impact on the results, because of feedbacks and non-linearities. I insist that you show the total SEMIC SMB somewhere (possibly in an appendix or supplement) so that the quality of the model can be judged.

> - p6 l18 Not clear what the shortcomings of the Krapp method to treat albedo were and neither how this has been improved for the present study. This requires some additional description. Extending on the last comment, changes to the albedo scheme likely also have an impact on the SMB and would lead to different tuning even for the same climate model input.

> We agree with the reviewer. We expand the section about the SEMIC model. In order to be consistent with parameters provided by Krapp et al. (2017) we switched back to the albedo scheme used by Krapp et al. (2017) for the new simulations.

So the improvement in the albedo scheme was not a very important improvement? As pointed out before, consistency may already be violated just by using a different climate model. Therefore, the consistency argument does not hold very strong for me.

> - p9 l25 These gradients were found as best fit to SMB simulated by a specific RCM (MAR) at different elevations. Applying these in your setup may be better than nothing, but for a consistent picture, these should ideally be recalculated based on your own model setup(SEMIC). Maybe, if you can run SEMIC at different elevation, you could get a feeling for the implied differences. *At the very least this inconsistency should be recognised and discussed as a shortcoming*.

This would be an interesting study. But for our application we follow the same argumentation above to the major point "parameter tuning". The parameters found by Edwards et al. (2014) are the most physical reliable and additionally we don't want to have different parameters between the three GCMs.

I agree with the argument that having different parameters for the different GCMs is not desirable and I see that it would be extra work to recalculate them with SEMIC. I completely disagree with the notion that the gradients are "the most physical reliable". These calculations have since been made with other models (e.g. Noël et al., 2016) with clearly different results, which shows that these parameters are model dependent and not unique solutions. I iterate my minimum requirement to mention in the text that the gradients are based on a different model setup and not consistent with the climate forcing applied for the projections.

- p11 l32 I am wondering in how far a detailed analysis of individual glaciers is justified given that an important aspect of the forcing in form of interaction with the ocean and sub-glacial hydrology is missing. The comparison suggests that we could hope to get the behaviour of individual glaciers in line with observations, which I consider very unlikely given the steady-state initialisation, coarse GCM-based forcing and lack of important forcing mechanisms.

This is indeed a good point raised. It is certainly true, that important forcing mechanisms like the oceanic forcing and subglacial hydrology are missing in this study, however, representing the dynamics of a glacier in the narrow fjords of Greenland well or representing the large NEGIS well, is only achieved with sufficient grid resolution and physics in the model, which our model both fulfils. This is indeed assessed by comparing individual glacier drainage basins with observation, like the surface velocity field. We are concerned about the statement 'given the steady-state initialisation' – we do not perform a steady-state initialisation at all, in contrast, we perform a complex initialisation procedure with mixture between inversion and

paleo-spin ups. This procedure has been the top procedure in an international benchmark assessing the ability of models to achieve a good initial state (Goelzer et al., 2018). The reviewer seems to have overlooked this substantial part of this study. The coarse GCM-based forcing is subsequently processed in SEMIC is improving the resolution and the anomaly forcing is making sure, that the SMB in individual glacier basins is in high resolution – so the glacier basins are forced on high resolution.

With steady-state initialisation I mean that the attempt is to bring the ice sheet to a steady state at 1960. The way this is done here, no transient dynamical processes are active at that point that arise from past climate forcing. In the absence of dedicated ocean forcing, the response in ice flow and outlet glaciers can only be based on SMB forcing from 1960 onwards (and possibly some unwanted model drift). I believe this is also what the second reviewer had in mind for his second general comment on the omitting of ice dynamics.

The claim to have the "top" procedure in the initMIP benchmark calls for some clarification. The model clearly achieves a very good match with the observed geometry. However, this is not the only factor that should be evaluated to judge the quality of an initialisation. It is specifically pointed out in the Goelzer et al. study that for this class of models, a better match with the observed geometry can be achieved by accepting a larger drift in the control experiment. The model drift in the control experiment of ISSM-AWI is the *largest* in the group of models with a similar initialisation method (data assimilation). Taking isolated results of an intercomparison out of context to falsely claim a superior modelling approach is inappropriate and should be avoided.

**General comments**

The validation presented in section 2.3 has important problems:

The correlation analysis shown in Figure 4 is not a meaningful validation. The year-to-year variability in the GCMs is not expected to coincided with that of RACMO, because the GCMs have their own internal variability. Correlation other than the long-term trend is pure coincidence, as can be seen from figure 3.

The comparison presented in Figure 5 is also meaningless, because the two SMB fields are by construction very similar. Panel a is RACMO(1990) and panel b is mean RACMO(1960-1990) + dSMB, where dSMB is by definition close to zero. The difference between the two is only due to inter-annual variability in RACMO and the GMCs, which again, are not expected to co-evolve.

In a first step, the absolute SMB of SEMIC for the different GCMs should be shown and compared to state-of-the-art RCM results.

Secondly, a meaningful validation of the SMB model in response to climate forcing is to force SEMIC with reanalysis data (e.g. ERA interim) and compare the resulting SMB with observations, RACMO or MAR. This is done for any other SMB model used for projections, whether in absolute mode or anomaly mode (e.g. Hanna et al., 2011, Fettweis et al., 2007, Noël et al., 2016, Vernon et al., 2013).

Afterwards it can be concluded that the absolute SMB is not ideal and the anomaly method can be applied.

A new "scenario" (without overshoot) has been added to the analysis. It is constructed by "cut-and-paste" based on the original RCP2.6 simulations of the individual GCMs. The procedure first identifies the time of overshoot for each individual GCM. Until this point the forcing remains the same as for the original scenario. From this point on an arbitrary 30 yr period from later in the individual simulations is repeated to fill the length of the original simulation. I see a number of problems with this ad hoc approach that are mainly in relation to model dependency that complicates the comparison between the GCMs.

First, the resulting forcing time series should be shown. I suspect they will show a step change of the forcing at the moment of overshoot. I am not sure how to interpret such forcing as it is unphysical. It is also highly model dependent, I suppose the resulting forcing should not be called a scenario for that exact reason.

The choice of 30 year period seems arbitrary. Why not use instead e.g. the last 30 or 50 years of the simulation?. The strong multi-decadal variability visible in the SMB time series suggests that a much longer time period would be appropriate. How robust are results to a different choice of the period?

There may be a fundamental problem with the constructed time series because of using the anomaly method. The reference period for the SMB anomaly is 1960-1990, so the forcing is calculated relative to that period. The temperature time series used to diagnose the time of overshoot is referenced to another time period. This implies an offset of the forcing in function of the global temperature mismatch between the GCMs over the 1960-1990 reference period. I am not sure re-referencing will solve the problem entirely, but it may be worth a try.

I am not sure addressing these points will be sufficient to make the taken approach look like a good idea. To revert to the original manuscript and removing the constructed forcing may be a viable option, too.

**Specific comments**

p1 l11 Clarify the use of scenarios in "for some scenarios". Probably you mean "for some models" or "for some experiments" if is any of the 3 models and 2 scenarios.

p1 l11-12 Why "most likely"? How do the different experiments differ in terms of the integrated SMB? Is there a clear difference between the runs that stabilise and those that continue to lose mass after 2300? Is SMB integrated in time or spatially? Clarify.

p1 l14 Do you mean SEMIC or the GCMs in "stem from the underlying climate model"? Clarify.

p1 l17 Delete "observed" after "observed"

p2 l3-4 "mass loss" is caused by "acceleration" and **decrease** in SMB.

p2 l6 Maybe omit "regional" since the study is not concerned with it.

p2 l16 Also here, clarify the use of "other scenarios". See comment p1 l11.

p2 l19 place "has exceeded 1.5C ..." before "and may exceed 4C by 2100".

p2 l25 remove two times "very" before "scarce" and before "extensive".

p3 l6 Reformulate. "most suitable" may be true in some cases, but is certainly not generally true.

p3 l19 Replace "volume" by "thickness".

p3 l19 Remove "surface".

p3 l29 Add after or replace "Numerical" by "thermo-mechanical".

p4 l12-13 add "M" after "melt rate" add "R" after "Refreezing" and adjust text below.

p5 l4 It is confusing that the analysis is done with 11 yr moving windows and the lines in the figures are plotted with 30 yr running mean. This makes it difficult to visually inspect the threshold criteria and the location of the dots seems off. Consider revising.

p5 l14 Reformulate "striking". A large scale average will always show less variability compared to a local region in a dynamic system.

p5 l25 Conservative interpolation may not be optimal for temperature, a quantity that physically cannot be conserved. I suspect that the imprint of the original GCM grid in the final product we see in Figure 11 and 12 may be related to that. For a vertically downscaled variable, I would not expect such a strong imprint.

p5 l26 Insert "on which SEMIC is run" after "0.05 grid", if that is correct.

p5 l29 Remove line break, still discussing downscaling.

p6 l7 Add "when MAR is used as forcing" after "best possible SMB and T_s fields".

p6 l13 Add a figure that shows the absolute SEMIC output, e.g. the 1960-1990 average given in rhs of equation 5.

p6 l20 The integral of deltaSMB in equation 5 from 1960 to 1990 should be zero and the integral of SMB_clim over the same period should be the same for all GCMs. It doesn't look like that in Fig 3, but maybe that is because of the running mean? To check!

p7 l17 Move "integrated" after SMB to avoid confusion between spatial and temporal integration.

P7 l2 How far are you from the ideal case? Please show that as a figure plotting the difference between the two reference SMBs.

p7 l26-29 This text is not part of the validation. Suggest to move to the results section.

p7 l30 This is not a meaningful validation. See general comments.

p8 l10 This is also not a meaningful validation. See general comments.

p9 l1 The causality of this sentence is not clear to me. Why could an arbitrary time period not be used if absolute SMB would be applied? I believe there may be a fundamental problem arising from the use of the anomaly method. See general comments.

p9 l3 What is the motivation for using the time period (2250-2280), or is that an arbitrary choice? Please clarify in the text.

p9 27 How do you apply observed velocities to land-terminating glaciers? Only at the ice front, as a boundary condition? Please clarify.

p9 l33 This compensation only applies at marine margins, I suppose. Clarify.

p10 l1 Maybe "have retreated".

p10 l20 Mention which version of BedMachine.

p10 l23 Clarify what noise is expected to be avoided.

p10 l25 I think it is safe to replace "125 kyr before 1990" by 125 kyr BP, with "(before present)" in first occurrence, to avoid the confusion between 1990 and 1960 in the following.

p10 l30 Consider discussing the temperature spinup with constant climate after relaxation at p10 l23 and adding it to Table 3.

p11 9-12 This could be mentioned before describing the method. In any case, reformulate. I don't think you really assume these statements to be true: Replace 'assumptions' by "simplifications". "The currently observed present-day elevation is taken constant for the entire glacial cycle". "the basal friction coefficients obtained from the inversion is taken constant for the past glacial cycle, and (3) the temperature changes from the GRIP record are applied to the whole ice sheet without spatial variations."

p11 l20 I agree that it may be a negligible effect. But where is the table with comparison of basal temperature against ice core results? The suggestion was not to remove the table, but to check the results that were presented in it.

p12 l27 The drift is ~15% of the magnitude of the lowest projection. That's not negligible. Replace "negligible" by "small".

p13 l7-9 Figure 8a and 8b are confused. Exchange.

p13 l9 Explain blue dots in Figure 8a.

P13 l9 The number in the table rounds to 400 m/a not 390 m/a.

p13 l11 Was the model run forward in time here or in the comparison, clarify.

p13 l13 What is the "assumed critical time"? Clarify.

p13 l17 Replace "negligible" by "small" and give a number.

p13 l20 Replace "variability" by "range". This is an ensemble range.

p13 l21 Name which figure (Fig 9) after "mass change".

p13 l22-23 Better discuss only SLE change, otherwise it gets confusing with the different sign of mass and SLE changes.

p13 l29 The comparison with Fig 1 is hampered by the different reference periods used for temperature and SMB. Consider producing a temperature time series re-referenced to the 1960-1990 reference period.

p14 l3-8 You could mention already here why the numbers are expected to be lower (missing ocean forcing, missing Greenland blocking in GCMs, ...). It seems more appropriate to compare against SMB-only results for this period instead.

p14 l9 Shouldn't some of this section 3.3 appear before the projections in chapter 3.2, since it shows the forcing that leads to the ice sheet results?

p14 l16 Replace "cooling" by "less warming", or are you comparing 2300 to 2100? Not always clear what we compare against.

p14 l20-22 Clarify this contradiction: "amplification is not well represented in MIROC5" <?> "with respect to the Arctic amplification phenomena the most plausible distribution of surface warming is produced by HadGEM2-ES and MIROC5".

p14 l21- p15 l5 The discussion of realism of future warming patterns remains very speculative and arbitrary.

p15 l2-5 The Watterson analysis is probably global, which may not be that meaningful for this Greenland application. This should be mentioned. MIROC5 often scores best when used with MAR compared to other GCMs. This could be mentioned, too.

p15 l21 Remove "increased" before thickening, or was it thickening already?

p15 l23 Not sure comparison with observations really holds here. You mean marginal thinning and central thickening as large-scale features? Yes, but the thinning must reach much further inland here. Consider revising.

p15 l25 Is there evidence from other studies that the 79 glacier is vulnerable? SMB changes in HadCM and MIROC seem very small here and the pattern of retreat looks almost identical. What happens in the unforced control run in this region? Are these changes in the figure also calculated relative to the control run (i.e. double differences)? They should! In any case it may be interesting to inspect and show the control run in a figure.

p15 l33-34 Here the discussion of 2100 and 2300 changes is mixed with recent changes.

p16 Maybe "ice discharge anomaly"?

p17 What is "It"? Clarify.

Figure 1 The offset between GCMS in temperature in (b) does not correspond with the curves in Fig 3 because of the different reference periods. This makes comparison later in the manuscript difficult and may also have an impact on the timing of the overshoot.

Figure 3 "according to Fig 4.". Replace "thick line" by "solid lines"

Figure 4 Remove or replace by a more meaningful comparison.

Figure 5 Remove or replace by a more meaningful comparison.

Figure 6 Are values trimmed at +-25? If yes, say so in the caption and give the min/max values.

Figure 8 Explain what the blue dots stand for in a. Does an r^2 =1.00 mean perfect correlation? How is that possible?

Figure 9 Replace "Straight" by "Solid"

Figure 10 May want to add an estimate for the SMB-only contribution.

**References:**

Fettweis, X. (2007). Reconstruction of the 1979-2006 Greenland ice sheet surface mass balance using the regional climate model MAR. The Cryosphere, 1(1), 21–40. http://doi.org/10.5194/tc-1-21-2007

Hanna, E., Huybrechts, P., Cappelen, J., Steffen, K., Bales, R. C., Burgess, E., et al. (2011). Greenland Ice Sheet surface mass balance 1870 to 2010 based on Twentieth Century Reanalysis, and links with global climate forcing. Journal of Geophysical Research, 116(D24), n/a–n/a. http://doi.org/10.1029/2011JD016387

Noël, B., Van De Berg, W. J., Machguth, H., Lhermitte, S., Howat, I., Fettweis, X., & Van Den Broeke, M. R. (2016). A daily, 1 km resolution data set of downscaled Greenland ice sheet surface mass balance (1958–2015). The Cryosphere, 10(5), 2361–2377. http://doi.org/10.5194/tc-10-2361-2016

Vernon, C. L., Bamber, J. L., Box, J. E., Van Den Broeke, M. R., Fettweis, X., Hanna, E., & Huybrechts, P. (2013). Surface mass balance model intercomparison for the Greenland ice sheet. The Cryosphere, 7(2), 599–614. http://doi.org/10.5194/tc-7-599-2013

---

## Referee Report (RR2)

I have now reviewed the paper by Rückamp et al. for a second time. In the first and second round of reviews major revisions were suggested by the reviewer(s). I have gone through the author's rebuttal letters and the revised manuscript and can say that the manuscript appears greatly improved to the previous round. All of the major scientific points raised by the reviewers have been appropriately addressed. I have only a couple of minor scientific points that I would like the authors to address (see list below). In addition, I would like to urge the authors to go through the manuscript with a fine brush to iron out any language lapses. I am listing below all I could find, but it is quite likely that I missed a few.

**Scientific comments**

1. On page 5 in the paragraph lines 16-22, you are saying that SEMIC provides annual mean surface temperatures. Are you using these temperatures as a Dirichlet boundary condition at the surface for the temperature equation or are you using the near-surface temperatures from the GCMs? If the former, how do you account for intra-annual temperature variabilities? If the latter, do you correct the temperature for the height difference between the GCM and ISSM? Please clarify!
2. I am also still missing what kind of boundary condition you apply at the base of the floating parts for the temperature equation. I suspect that no matter what you specify there, it wouldn't affect your results much, but for completion it would be nice if you could add this piece of information to the second paragraph on page 10 (I think I would set a Dirichlet condition to the local pressure melting point temperature).
3. Could you add somewhere in the inversion section, if you use any sort of regularisation (e.g. Tikhonov?) in your basal friction coefficient inversion? If so, how do you estimate the optimal parameters for the regularisation (usually done with an L-curve analysis but this might be a bit expensive for your advanced initialisation method)? If you do not use regularization, how to you ensure that you get a smooth velocity field?

**Specific comments**

I am noting all typos and grammar issues I found but the authors need to go through the manuscript with a fine brush and give it a good workover in terms of language style in general. Also I am not sure what the ESD policy is on whether North American or British spelling should be used, but in this manuscript it is mostly a strange mixture e.g. coloured (BE), but initialized (AE).

Throughout the manuscript it is **sea level**, but **sea-level rise**.

P1L5 approach between = approach of

P1L7 Just to check, are citations allowed in the abstract? I don't find it necessary here as you cite Krapp et al. 2017 later in the text.

P1L13 Here and throughout spatial-integrated=spatially-integrated

P1L12-14 please replace "never falls below zero" with something along the lines of "SMB remains positive". Also "a recovery of SMB towards values of slightly below present day" is a bit to general for my taste. You have to know that the SMB is positive for Greenland to understand this. This is OK for the main paper where you give numbers, but if the SMB for the whole ice sheet was negative, this sentence would sound strange. Why instead not write that SMB "decreases in the latter half of the simulation period reaching values similar to present day"?

P2L1 This sentence makes no sense. Are you trying to say that Greeland's contribution to global sea-level rise has been 20%? Please rephrase.

P2L2 delete "to global sea-level rise"

P2L4 Nerem et al. 2018 show only that SLR is accelerating, but not why. Please use a different reference.

P2L6 keep a global = keep the global

P2L19 by that time = by 2100

P2L19 by 2000 and … should that not read 2100?

P2L26 delete "abated"

P3L6 add at the end: ", which is employed/used in this study."

P3L6-7 delete "by means"

P3L16 make sure you introduce the acronyms at first use. You don't seem to introduce them in the abstract. Please make sure this conforms with the ESD guidelines.

P3L30 delete "very"

P4L1 based on reconstruction. What reconstruction?

P4L10 parentheses around SMB missing

P4L16 from available energy = from the available energy

P4L31-P5L1 MIROC5 also exceeds this threshold according to Figure 1

P5L3 it stabilizes= temperatures stabilize

P5L9 delete second "the"

P6L8 parameter=parameters

P6L13 delete "However"

P6L18 still lacks details and quality compared = still lack details compared

P6L21 differs up to = differs by up to

P6L21 full stop after 200 Gt $a^{-1}$

P7L20-21 Delete sentence starting with "This one way coupling …". You have said this before.

P7L28 delete focusing on negative emissions

P7L30-31 Delete sentence starting with "As mentioned before …". You have said this before.

P8L13 delete "over the year from SMB"

P8L22 below the magnitude of present day = present day magnitude

P8L27 among the models is rather similar = is rather similar among the models

P8L29 agrees well to = agrees well with

P8L31 delete sentence. Try to avoid hollow statements like this.

P8L32 change sentence structure to read: "For the available RACMO2.3 time series and the SEMIC-GCMs we have computed the interannual SMB variability.

P9L7 year missing from citation

P11L13 please write out approximately here and throughout

P11L18 remove parentheses around Rignot and Mouginot citation

P11L20-24 Out of curiosity, can you restart ISSM simulations from different 3D unstructured grids? Is there a routine that interpolates these fields between the meshes?

P13L16 I am not a native speaker, but I don't think I would capitalise compass directions unless I talk about a specific region e.g. Western Rockies. But I leave this up to the editor.

P14L11 less=lower

P14L23 experienced = experiences

P14L24 change to something like "mass loss from a decline/less positive surface mass balance"

P16L12 change to "response is different across all models"

P16L16 delete "abated"

P17L12 whether=neither

P18L12 change to "Despite all three GCMs being based on …."

P18L13 instead of considerably large, please give a number and avoid such vague language.

**Figures**

Figure 4 Is there a solid black line in the upper panel? Might be worth it to bring the black line to the front in both panels.

Figure 5 alls = all.

The thick solid lines show 30-year moving means. (all lines are solid in this figure!)

Figures 7 & 8 Why are the fields from IPSL so patchy? Is this a resolution issue?

Figure 9 In caption: how can this be observed velocities if HadGEM2-ES was used? Can we also have different colourbar labels? They look odd.

Figure 12 Caption change to: The solid black line indicates …

Best wishes,
Clemens Schannwell, Geosciences, University of Tübingen, Germany

---

## Author Response (AR2)

Dear Editor, dear Reviewer,

the authors wish to thank again the reviewer for the detailed review containing many helpful remarks and constructive criticism.

From the review below we have identified, that you have still these major concerns:

  1) validation of SMB forcing

  2) the modified RCP2.6 scenario without overshoot

  3) reference times for SMB forcing and temperature time series

As these concerns appear at several locations in the review, we will answer to them at one place marked with "Answer to MC1,2,3,". At the other locations, we will refer to this comment.

We think, that our wording to refer to GCM's also when we present and discuss output of SEMIC driven by GCMs might have lead to serious confusion for which we apologize. In order to distinguish better in the current version, we have introduced e.g. SEMIC-HadGEM in order to denote the resulting fields from SEMIC. We hope that this helps to dissolve this issue for future readers.

Further technicalities: below we answer each point raised by the reviewer and mark our answer in green colour. Our point-to-point answers to the previews review and where kept by the reviewer and also parts of the first reviews are included, so we kept them here too. Every text from a previous review is in italic, either reviewers in black or our response in blue. 'Done.' denotes that this point is solved in the revised version of the manuscript. This could be that it will be either done directly, or that due to other changes the point does not arise any more, or that the point has been answered at another place in this text already.

We recognise that our manuscript is/was very labour intensive to review and we want to thank sincerely reviewer 1 and the editor for all the time and effort spent to improve the manuscript. In the course of revising the manuscript we spend a lot of time discussing the view of climate modellers on our way of dealing with GCM output, as well as ice flow modellers perspective of forcing ice models. We can say that we learnt a lot from these discussions, which only appeared due to the critics raised in the review. Many thanks!

Best wishes,

Martin & Co-authors

Reviewer 1

**Summary**

I am evaluating this paper for the second time after major revisions in the first round. The manuscript has changed considerably since the last iteration, has improved in response to the reviewer's comments and has seen new material being added. However, some major issues remain or have been introduced with the modifications, as is the case for the validation part and the "scenarios without overshoot". The language also needs further improvements to clean up errors and make the text clearer.

I will first respond to some of the author comments (blue), with my initial comments indented twice. General and specific comments on the new version follow below.

**Response to the discussion**

*Yes, indeed this is an important point and we followed the reviewers suggestion. With using the parameters of Krapp et al. (2017) the direct output of the SMB from SEMIC has a misfit of about ~2m/a and a correlation of ~$r^2$=0.5 by comparing SMB_RACMO_1960-1990 and dSMB_SEMIC_1960-1990 (almost similar for all GCMs used).*

This evaluation should be extended and backed with figures to appear also in the manuscript. See general comments.

See answer MC1.

*However, recalling Equation 3 and 4 from the manuscript, we do not use the direct output of SEMIC, but apply anomalies computed using SEMIC. The benefit of our approach is, that only the GCM trends of SMB changes are added to the RACMO SMB reference field, which represents the real SMB distribution very well. If we compare the computed SMB to RACMO (according to Eq. 3 and 4 without the synthetic SMBcorr), for instance for the HadGEM2-ES year 1990, it shows a very good agreement (Figure 2). See also answer to specific comment "p10 l2" below. In the revised manuscript we dedicate an own section to this issue.*

This comparison does not really validate the SEMIC model. It only shows that at a given point (year 1990) the anomalies of SEMIC are close to zero. See general comments.

See answer MC1.

*We expand the section about the SEMIC model in order to give the reader a better understanding of the model. In the new version of the manuscript we also review in the introduction section briefly the already existing alternatives used and relate the discussion section accordingly. The reason we have not included too much detail on that issue previously is, that we basically apply SEMIC and that the model in itself and all the parameter tuning is work done by Krapp et al., 2017. The advantage of using a semi-complexity model is indeed its simplicity and cost efficiency, which would allow ice sheet modellers to also run computation up to time scales of thousands of years (e.g. until 5000) studying long-term commitment of various emission scenarios and hence not be limited by the availability of regional climate model output.*

Multi-millennial simulations are not relevant for the current study. The question is if the chosen model is an appropriate tool for the presented type of simulation. The SEMIC model has so far not been validated for the use with GCM input data, the way you are using it for the projections. Showing a proper validation is the price you have to pay for that novelty.

See answer MC1.

*The authors rely on the parameter settings of the SEMIC model, which have been optimised for a different climate model input (Krapp et al., 2017). The Krapp et al. study shows that the SEMIC model can well approximate the MAR SMB results given MAR climate input. It must however be expected that the parameters that were chosen for a completely different climate input (different model, RCM vs GCM) are not optimal. Unless evidence can be provided that the applied parameters are indeed suited for the GCM forcing used in the present study, the model parameters should be optimised. Discussion on differences to other results (e.g. as done compared to Fürst et al., 2015) hinges on the implied sensitivity of the SMB model, which is currently not possible to be judged.*

*We haven chosen the same parameters of SEMIC as Krapp et al., 2017, due to the following reason: the*

*parameter tuning procedure performed by Krapp et al., 2017 aimed to find a parameter set which gives a best fit between SMB and skin temperature Ts of SEMIC with only a limited number of processes and simpler parameterisations than a regional climate model with full complexity would derive. As a regional climate model is typically validated against reanalysis data and observations, the best match between SMB and Ts of SEMIC and regional climate model (in that case MAR) is the best way to represent the processes and their parameters in SEMIC. We see it thus as a tuning of the parameterisation of the processes. Once the process description in SEMIC is optimised, any type of input, either GCM or reanalysis data fields, will lead to the best possible SMB and Ts fields that SEMIC can produce. Still, the GCM will lack the best atmospheric fields over the ice sheet, as it is limited in resolution compared to a regional climate model. Given experiences we made from these three GCMs used in this study, which are all have different drawbacks, which would mean to have a tuning for each of them and this tuning would then make the whole benefit of having a semi-complexity model with low costs meaningless. Furthermore, it would basically mean to compensate far too low near surface temperatures with SEMIC parameters, which would offset the whole comparison of GCM forcing. Therefore, we have chosen a different approach: we compensate for this by using the SEMIC output only as an anomaly.*

I understand the argument to avoid tuning the model for individual GCMs and I agree with the point on compensating errors in the GCM. Nevertheless, I think you will agree that if you were to tune SEMIC for another RCM (say RACMO or HIRHAM), you would end up with different parameters. I believe it is important to recognise that, even if you chose to do nothing about it and use the MAR based parameters.

Yes, we agree, tuning SEMIC with another RCM will likely change the parameters, but we can only speculate to which extent. We followed the reviewer's recommendation for the comment p6 l7. Also, we added to the paragraph 2.2 (p6 l7): "If SEMIC is tuned with another RCM (e.g. RACMO or HIRHAM), the parameters will be different."

The situation here is worse, because using forcing from a GCM implies different characteristics, like smoother gradients and less resolved geometry compared to the RCM. It is possible that these characteristic differences between RCM and GCM (not individual model bias) have an important impact on the modelled SMB. I believe it is your responsibility to show that the parameters that you are using are indeed appropriate for the given purpose. You should show how the absolute SMB looks like for the different GCMs and compare that to reconstructions and/or state-of-the-art RCM results.

I agree that using the anomaly method is a good choice, as it circumvents *some* of the biases in the absolute SMB products you are producing with SEMIC. Nevertheless, the reference SMB has an important impact on the results, because of feedbacks and non- linearities. I insist that you show the total SEMIC SMB somewhere (possibly in an appendix or supplement) so that the quality of the model can be judged.

See Answer to MC1.

*p6 l18 Not clear what the shortcomings of the Krapp method to treat albedo were and neither how this has been improved for the present study. This requires some additional description. Extending on the last comment, changes to the albedo scheme likely also have an impact on the SMB and would lead to different tuning even for the same climate model input.*

*We agree with the reviewer. We expand the section about the SEMIC model. In order to be consistent with parameters provided by Krapp et al. (2017) we switched back to the albedo scheme used by Krapp et al. (2017) for the new simulations.*

So the improvement in the albedo scheme was not a very important improvement? As pointed out before, consistency may already be violated just by using a different climate model. Therefore, the consistency argument does not hold very strong for me.

For the first version of the manuscript, the resulting albedo was more plausible by using our "improved" albedo scheme compared to the scheme used by Krapp et al. (2017). But this was only due to the application error (mentioned in the preamble of our response in the first round of the review).

*p9 l25 These gradients were found as best fit to SMB simulated by a specific RCM (MAR) at different elevations. Applying these in your setup may be better than nothing, but for a consistent picture, these should ideally be recalculated based on your own model setup(SEMIC). Maybe, if you can run SEMIC at different elevation, you could get a feeling for the implied differences. \*At the very least this inconsistency*

*should be recognised and discussed as a shortcoming*.

*This would be an interesting study. But for our application we follow the same argumentation above to the major point "parameter tuning". The parameters found by Edwards et al. (2014) are the most physical reliable and additionally we don't want to have different parameters between the three GCMs.*

I agree with the argument that having different parameters for the different GCMs is not desirable and I see that it would be extra work to recalculate them with SEMIC. I completely disagree with the notion that the gradients are "the most physical reliable". These calculations have since been made with other models (e.g. Noël et al., 2016) with clearly different results, which shows that these parameters are model dependent and not unique solutions. I iterate my minimum requirement to mention in the text that the gradients are based on a different model setup and not consistent with the climate forcing applied for the projections.

We agree that "most physical reliable" is misleading here. Also, the intended sentence about this issue doesn't make it into the revised version of the manuscript for which we apologize. Now, we have added: "This relationship was estimated from a set of MAR simulations in which the ice sheet surface elevation was altered. [...] Please note, that the employed relationship with their parameters may change using a setup from SEMIC."

*p11 l32 I am wondering in how far a detailed analysis of individual glaciers is justified given that an important aspect of the forcing in form of interaction with the ocean and sub-glacial hydrology is missing. The comparison suggests that we could hope to get the behaviour of individual glaciers in line with observations, which I consider very unlikely given the steady-state initialisation, coarse GCM-based forcing and lack of important forcing mechanisms.*

*This is indeed a good point raised. It is certainly true, that important forcing mechanisms like the oceanic forcing and subglacial hydrology are missing in this study, however, representing the dynamics of a glacier in the narrow fjords of Greenland well or representing the large NEGIS well, is only achieved with sufficient grid resolution and physics in the model, which our model both fulfils. This is indeed assessed by comparing individual glacier drainage basins with observation, like the surface velocity field. We are concerned about the statement 'given the steady-state initialisation' – we do not perform a steady-state initialisation at all, in contrast, we perform a complex initialisation procedure with mixture between inversion and paleo-spin ups. This procedure has been the top procedure in an international benchmark assessing the ability of models to achieve a good initial state (Goelzer et al., 2018). The reviewer seems to have overlooked this substantial part of this study. The coarse GCM-based forcing is subsequently processed in SEMIC is improving the resolution and the anomaly forcing is making sure, that the SMB in individual glacier basins is in high resolution – so the glacier basins are forced on high resolution.*

With steady-state initialisation I mean that the attempt is to bring the ice sheet to a steady state at 1960. The way this is done here, no transient dynamical processes are active at that point that arise from past climate forcing. In the absence of dedicated ocean forcing, the response in ice flow and outlet glaciers can only be based on SMB forcing from 1960 onwards (and possibly some unwanted model drift). I believe this is also what the second reviewer had in mind for his second general comment on the omitting of ice dynamics.

The dynamical processes of past climate forcing are included in the way that the ice flow model is forced with it, but within the spin-up and inversion sequences, not the full geometric evolution is allowed, as this is (for all ice models) leading to initial states that are far from present day geometry. The response to past climate forcing is included, but not to the extent that all ice modellers would like to include it. As we stated before, the projections are based on SMB forcing, as the ocean forcing is kept fixed. This is, however, clearly stated in the manuscript and is also not such uncommon in ice flow model projection studies.

The claim to have the "top" procedure in the initMIP benchmark calls for some clarification. The model clearly achieves a very good match with the observed geometry. However, this is not the only factor that should be evaluated to judge the quality of an initialisation. It is specifically pointed out in the Goelzer et al. study that for this class of models, a better match with the observed geometry can be achieved by accepting a larger drift in the control experiment. The model drift in the control experiment of ISSM-AWI is the *largest* in the group of models with a similar initialisation method (data assimilation). Taking isolated results of an intercomparison out of context to falsely claim a superior modelling approach is

inappropriate and should be avoided.

With "top procedure" we refer here to a method that brings both initialization methods together: Long interglacial temperature spin-up and data-inversion. It should not be understand as the method of choice. And indeed, the drift is larger compared to similar initialization methods (inversion). But some of these models (e.g. JPL-ISSM) perform a relaxation run that is much larger than for the AWI-ISSM model (50.000 years compared to 50 years, respectively). For those models it is expected, that the drift is lower.

**General comments**

The validation presented in section 2.3 has important problems:

The correlation analysis shown in Figure 4 is not a meaningful validation. The year-to-year variability in the GCMs is not expected to coincided with that of RACMO, because the GCMs have their own internal variability. Correlation other than the long-term trend is pure coincidence, as can be seen from figure 3.

The comparison presented in Figure 5 is also meaningless, because the two SMB fields are by construction very similar. Panel a is RACMO(1990) and panel b is mean RACMO(1960-1990) + dSMB, where dSMB is by definition close to zero. The difference between the two is only due to inter-annual variability in RACMO and the GMCs, which again, are not expected to co-evolve.

In a first step, the absolute SMB of SEMIC for the different GCMs should be shown and compared to state-of-the-art RCM results.

Secondly, a meaningful validation of the SMB model in response to climate forcing is to force SEMIC with reanalysis data (e.g. ERA interim) and compare the resulting SMB with observations, RACMO or MAR. This is done for any other SMB model used for projections, whether in absolute mode or anomaly mode (e.g. Hanna et al., 2011, Fettweis et al., 2007, Noël et al., 2016, Vernon et al., 2013).

Afterwards it can be concluded that the absolute SMB is not ideal and the anomaly method can be applied.

Answer to MC1:

We are afraid that here a misunderstanding is arising. One thing is to validate the SMB model itself, meaning the way the energy balance is computed and the parameters that are tuned by Krapp et al. (2017). Another thing is to validate the SMB fields that are computed. A real validation is tricky, and is beyond the scope of this study. What we do instead is to assess how plausible the fields are. In this respect we compare them with RACMO in the same time period and check if both the pattern is matching reasonably and to check if the interannual variability is similar, meaning not exactly every year matching the total SMB, but to assess if years of extreme SMB are present in the SMB fields we have obtained using SEMIC and if the frequence matches reasonably with RACMO.

Additionally, we have forced SEMIC with ERA-Interim as requested and compare it with the absolute fields derived by SEMIC forced with the GCM data and RACMO2.3 (Figure below). Shown is a time average from 1979 to 2012. The spatial-integrated SMB for HadGEM2, IPSL, MIROC5, ERA-Interim and RACMO is 535, 556, 480, 143 and 351 Gt/a, respectively. Though the general large-scale patterns among all of these fields agree fairly well, there is an offset between the spatial-integrated SMB.

[Figure]

|a) SEMIC-HadGEM2|b) SEMIC-IPSL|c) SEMIC-MIROC5|d) SEMIC-ERA-Interim|e) RACMO2.3|

In order to assess the interannual variability, we show the time series from 1979-2012 of the spatial-averaged SMB for all of SMB fields (Figure below). The interannual variability is with respect to the frequency and magnitude in a good agreement. RACMO2.3 and ERA-Interim are basically coherent, which is not surprising as RACMO2.3 is forced with ERA-Interim.

We will present a comparison between RACMO and the absolute SMB field derived by forcing SEMIC with HadGEM2 for the reference time period of 1960-1990 (Fig. 3 in the new version of the manuscript).

[Figure]

Additionally, a new figure of calculated variations between consecutive years demonstrates the interannual variability (Fig. 4 in the new version of the manuscript). Therefore, the old Figs. 4 and 5 in the manuscript are replaced by new figures.

The comparison we have made in the last version of the manuscript should not be understand as validating historical with reanalysis results. We intended to show, that our SMB forcing matches well the RACMO time series (as we thought this is what the reviewer requested). However, according to that, the choice of the year 1990 was not appropriate as ΔSMB is by definition small. However, replacing this exemplary 1990-analysis (Fig. 5) with the year 2016, the latest we have from the RACMO2.3 time, series where ΔSMB is not small, still shows a very good agreement (as you can see from the statistical values in Fig. 4). However, as mentioned above, this analysis is dropped in the new version of the manuscript and replaced by new figures.

A new "scenario" (without overshoot) has been added to the analysis. It is constructed by "cut-and-paste" based on the original RCP2.6 simulations of the individual GCMs. The procedure first identifies the time of overshoot for each individual GCM. Until this point the forcing remains the same as for the original scenario. From this point on an arbitrary 30 yr period from later in the individual simulations is repeated to

fill the length of the original simulation. I see a number of problems with this ad hoc approach that are mainly in relation to model dependency that complicates the comparison between the GCMs.

First, the resulting forcing time series should be shown. I suspect they will show a step change of the forcing at the moment of overshoot. I am not sure how to interpret such forcing as it is unphysical. It is also highly model dependent, I suppose the resulting forcing should not be called a scenario for that exact reason.

The choice of 30 year period seems arbitrary. Why not use instead e.g. the last 30 or 50 years of the simulation?. The strong multi-decadal variability visible in the SMB time series suggests that a much longer time period would be appropriate. How robust are results to a different choice of the period?

Answer to MC2:
We agree that the resulting forcing should be shown. The corresponding figure for the RCP2.6 scenario without overshoot is shown below. This figure is added to the new version of the manuscript as Fig. 5b.

There occurs no step change in the SMB time series. Indeed, the choice of the years from 2250-2280 is - to some extent - arbitrarily. The only restrictions that we made to the reused time period is a stabilized climate close to 1.5°C warming (in terms of temperature change, i.e. no long term decrease or increase in temperature) and to cover a time period with decadal variability to account for extreme years. Of course, we could use the last 30 or 50 years, but that will not change the results significantly. The frequency and amplitudes of the temperature change peaks (or increase/decrease of SMB) do not change too much. We have used different time periods of stagnant climate and the results (i.e. sea level contribution) are basically similar. In contrast, we have run the simulations with a reused time period around the "time of overshoot" where the warming still occurs. These simulations lead to more mass loss compared to the scenarios with overshoot. This also mentioned in the manuscript (p17 l12). This is due to the fact, that one would prolong the scenario with data of strongest climate change.

We have aimed to improve the description of the construction of this scenario in order to clarify the points raised in the review.

[Figure]

There may be a fundamental problem with the constructed time series because of using the anomaly method. The reference period for the SMB anomaly is 1960-1990, so the forcing is calculated relative to that period. The temperature time series used to diagnose the time of overshoot is referenced to another time period. This implies an offset of the forcing in function of the global temperature mismatch between the GCMs over the 1960-1990 reference period. I am not sure re-referencing will solve the problem

entirely, but it may be worth a try.

Answer to MC3:

First of all, we will not re-reference the temperature time series or re-analyze the diagnosed times of overshoot as the warming in the Paris Agreement is clearly defined as "warming above pre-industrial levels". However, using a time period from 1960-1990 as the reference period for determining the time of overshoot will shift those for HadGEM2-ES from 2023 to 2024, for IPSL-CM5A-LR from 2009 to 2011 and for MIROC5 from 2043 to 2050. We think you will agree, that this will not change the SMB forcing for the constructed SMB for the RCP2.6 without overshoot dramatically. The shift is largest for MIROC5 but this is due to the fact that the warming is more gently and with that the decline in SMB.

The other way around, providing the SMB forcing relative to the same reference period as the temperature curve (Fig. 1) is unfeasible as the RACMO dataset is not available for the pre-industrial time period (1661-1880).

However, we think that the reference periods do not pose a major problem. The ΔSMB (Eq. 5) is per definition close to zero for the pre-industrial or the historical (approx. <1990) runs. So for any selected reference period (approx. <1990) we would end-up with more or less equal numbers for the reference SMBs from the GCMs and for the background SMB field (here SMB_RACMO(1960-1990)). Consequently - recall Eq. 4 - the course of the SMB forcing based on the original RCP2.6 scenario will not change but may be offset by a few Gt (but this would be buffered in the end by the imposed SMB_correction (Eq. 8)). The construction of the scenario without overshoot is therefore independent of the selected SMB reference period. The trends of the warming (Fig. 1) and the SMB forcing (Fig. 3) are therefore consistent and can be compared directly.

I am not sure addressing these points will be sufficient to make the taken approach look like a good idea. To revert to the original manuscript and removing the constructed forcing may be a viable option, too.

See answer to MC2 and MC3.

**Specific comments**

p1 l11 Clarify the use of scenarios in "for some scenarios". Probably you mean "for some models" or "for some experiments" if is any of the 3 models and 2 scenarios.

Done.

p1 l11-12 Why "most likely"? How do the different experiments differ in terms of the integrated SMB? Is there a clear difference between the runs that stabilise and those that continue to lose mass after 2300? Is SMB integrated in time or spatially? Clarify.

We dropped most likely. In the whole manuscript, integrated SMB refers to a spatially integrated SMB. In the revised version of the manuscript, we have added "spatial" or "time" where appropriate. The differences between spatial-integrated SMB can be seen in Fig. 3. None of the curves fall below zero (except for a few individual years).

p1 l14 Do you mean SEMIC or the GCMs in "stem from the underlying climate model"? Clarify.

Done. We mean GCM.

p1 l17 Delete "observed" after "observed"

Done

p2 l3-4 "mass loss" is caused by "acceleration" and decrease in SMB.

Done.

p2 l6 Maybe omit "regional" since the study is not concerned with it.

Done.

p2 l16 Also here, clarify the use of "other scenarios". See comment p1 l11.

Done.

p2 l19 place "has exceeded 1.5C ..." before "and may exceed 4C by 2100".

Done.

p2 l25 remove two times "very" before "scarce" and before "extensive".

Done.

p3 l6 Reformulate. "most suitable" may be true in some cases, but is certainly not generally true.

Done. We replaced "suitable" with "efficient".

p3 l19 Replace "volume" by "thickness".

Done.

p3 l19 Remove "surface".

Done.

p3 l29 Add after or replace "Numerical" by "thermo-mechanical".

Done.

p4 l12-13 add "M" after "melt rate" add "R" after "Refreezing" and adjust text below.

Done.

p5 l4 It is confusing that the analysis is done with 11 yr moving windows and the lines in the figures are plotted with 30 yr running mean. This makes it difficult to visually inspect the threshold criteria and the location of the dots seems off. Consider revising.

Done. We make the analysis now in a 30-yr moving window and use the 30-yr moving window consistently in the manuscript (Except for the new Fig 5b, where we used for illustration a moving window of 15 years, otherwise we get a constant line for the repeated 30-yr time period). This changed the numbers slightly (HadGEM2-ES from 2021 to 2023 and IPSL-CM5A from 2041 to 2043; MIRCO5 remains at 2009). We have updated the numbers in the text and figures accordingly.

p5 l14 Reformulate "striking". A large scale average will always show less variability compared to a local region in a dynamic system.

Done. We have deleted this sentence.

p5 l25 Conservative interpolation may not be optimal for temperature, a quantity that physically cannot be conserved. I suspect that the imprint of the original GCM grid in the final product we see in Figure 11 and 12 may be related to that. For a vertically downscaled variable, I would not expect such a strong imprint.

Indeed, we have rerun SEMIC with updated input data using bilinear interpolation. This smoothed out the imprint (at least for HadGEM2 and MIROC5; IPSL still shows the imprint). By the way, we found an error for the fields in the Fig. 12 (lower panels). We computed the difference between 2300 and 2100.

p5 l26 Insert "on which SEMIC is run" after "0.05 grid", if that is correct.

Done. This is correct.

p5 l29 Remove line break, still discussing downscaling.

Done.

p6 l7 Add "when MAR is used as forcing" after "best possible SMB and T_s fields".

Done.

p6 l13 Add a figure that shows the absolute SEMIC output, e.g. the 1960-1990 average given in rhs of equation 5.

See answer to MC1.

p6 l20 The integral of deltaSMB in equation 5 from 1960 to 1990 should be zero and the integral of SMB_clim over the same period should be the same for all GCMs. It doesn't look like that in Fig 3, but maybe that is because of the running mean? To check!

We have checked that. The integral from 1960 to 1990 for deltaSMB is zero and SMB_clim is for all 3 GCMs 12001Gt.

p7 l17 Move "integrated" after SMB to avoid confusion between spatial and temporal integration.

Done.

P7 l2 How far are you from the ideal case? Please show that as a figure plotting the difference between the two reference SMBs.

We think, this is not necessary. In a new figure (see answer to MC1) we show the RACMO reference field and the SEMIC-HadGEM2-ES reference field.

p7 l26-29 This text is not part of the validation. Suggest to move to the results section.

As we have rewritten this section and partly restructured this paragraph, this sentence still appear here.

p7 l30 This is not a meaningful validation. See general comments.

See answer to MC1.

p8 l10 This is also not a meaningful validation. See general comments.

See answer to MC1.

p9 l1 The causality of this sentence is not clear to me. Why could an arbitrary time period not be used if absolute SMB would be applied? I believe there may be a fundamental problem arising from the use of the anomaly method. See general comments.

You are right, absolute fields could also be applied. See also answer to MC2.

p9 l3 What is the motivation for using the time period (2250-2280), or is that an arbitrary choice? Please clarify in the text.

See answer to MC2.

p9 27 How do you apply observed velocities to land-terminating glaciers? Only at the ice front, as a boundary condition? Please clarify.

We add "at the ice front".

p9 l33 This compensation only applies at marine margins, I suppose. Clarify.

No, the compensation applies at marine and land terminating margins. But at land terminating margins melting is compensated. We rewrite to ":... calving and melting exactly compensates …".

p10 l1 Maybe "have retreated".

Done.

p10 l20 Mention which version of BedMachine.

Done. We use here version 2.

p10 l23 Clarify what noise is expected to be avoided.

Done. We rewrite to  "… to avoid spurious noise that arise from errors and biases in the datasets".

p10 l25 I think it is safe to replace "125 kyr before 1990" by 125 kyr BP, with "(before present)" in first occurrence, to avoid the confusion between 1990 and 1960 in the following.

Done.

p10 l30 Consider discussing the temperature spinup with constant climate after relaxation at p10 l23 and adding it to Table 3.

We think this is not necessary, as it is not relevant for the paper. The spinup with constant climate is just used as a best guess for the initial value for the paleo spinup. See also comment to p11 l20.

p11 9-12 This could be mentioned before describing the method. In any case, reformulate. I don't think you really assume these statements to be true: Replace 'assumptions' by "simplifications". "The currently observed present-day elevation is taken constant for the entire glacial cycle". "the basal friction coefficients obtained from the inversion is taken constant for the past glacial cycle, and (3) the temperature changes from the GRIP record are applied to the whole ice sheet without spatial variations."

Done. The paragraph is moved to the beginning of the initialization approach description and slightly rewritten.

p11 l20 I agree that it may be a negligible effect. But where is the table with comparison of basal temperature against ice core results? The suggestion was not to remove the table, but to check the results that were presented in it.

We have decided to drop the table from the manuscript as it is not relevant for this paper. We are currently working on a manuscript that will present the hybrid spin-up approach in detail.

p12 l27 The drift is ~15% of the magnitude of the lowest projection. That's not negligible. Replace "negligible" by "small".

Done. However, the 15% is calculated to scenario that has almost no change in SMB. Compared to all other scenarios the drift is negligible.

p13 l7-9 Figure 8a and 8b are confused. Exchange.

Done.

p13 l9 Explain blue dots in Figure 8a.

Done. Blue dots represent floating points.

P13 l9 The number in the table rounds to 400 m/a not 390 m/a.

Done.

p13 l11 Was the model run forward in time here or in the comparison, clarify.

Done. We have the sentence rewritten to: The analysis here was done on the original native grid with the high resolution in fast flow regions and on the model was already run forward in time. Compared to this values, the AWI-ISSM results on the regular 5km grid given in Goelzer et al (2017) have a lower RMS value of <20ma^-1.

p13 l13 What is the "assumed critical time"? Clarify.
The "critical time" was introduced in the chapter 2.3 (p8 l7) as "The year 1997 was identified as the critical time of Greenland's peripheral glaciers and ice caps mass balance decrease." The "critical time" was then again used in chapter 2.6 (p10 l31). So, we don't change anything here.

p13 l17 Replace "negligible" by "small" and give a number.

Done. The numbers are added "... (-1.4 and -0.7mm for 2100 and 2300, respectively).".

p13 l20 Replace "variability" by "range". This is an ensemble range.

Done.

p13 l21 Name which figure (Fig 9) after "mass change".

Done.

p13 l22-23 Better discuss only SLE change, otherwise it gets confusing with the different sign of mass and SLE changes.

Done.

p13 l29 The comparison with Fig 1 is hampered by the different reference periods used for temperature and SMB. Consider producing a temperature time series re-referenced to the 1960-1990 reference period.

See answer to MC3.

p14 l3-8 You could mention already here why the numbers are expected to be lower (missing ocean forcing, missing Greenland blocking in GCMs, ...). It seems more appropriate to compare against SMB-only results for this period instead.

Done. In the updated manuscript we show additionally the sea level contribution of 0.4 mm/a from a SMB result (RACMO2.3) and have adjusted the text accordingly: "Since a future ocean forcing and calving front retreat is not considered here, the response of the ice sheet is likely underestimated here.

Comparing the sea level contributions of each GCM to the sea level contribution of 0.4 mm/a calculated from RACMO2.3 for the same period (dashed black line in Fig. 11) reveals a better agreement. HadGEM2-ES reaches this value 8 years later for RCP2.6 with overshoot and 9 years later for the RCP2.6 scenario without overshoot; IPSL-CM5A 10 years later for RCP2.6 with overshoot."

p14 l9 Shouldn't some of this section 3.3 appear before the projections in chapter 3.2, since it shows the forcing that leads to the ice sheet results?

Done. We have moved this section as suggested.

p14 l16 Replace "cooling" by "less warming", or are you comparing 2300 to 2100? Not always clear what we compare against.

Done. We compare here 2300 with 2100. We have rewritten this sentence.

p14 l20-22 Clarify this contradiction: "amplification is not well represented in MIROC5" <?> "with respect to the Arctic amplification phenomena the most plausible distribution of surface warming is produced by HadGEM2-ES and MIROC5".

Done. The paragraph is rewritten: "Although we do not have a measure to judge future climate warming trends, but with respect to the Arctic amplification phenomena the most plausible distribution and magnitude of surface warming is produced by HadGEM2. By contrast, MIROC5 produces less pronounced warming over Greenland that is similar to the global mean warming but exhibits a plausible pattern of warming. IPSL is spatially and temporally experiencing the largest warming; however, the distribution is not in agreement with the Arctic amplification."

p14 l21- p15 l5 The discussion of realism of future warming patterns remains very speculative and arbitrary.

Yes, we agree. But the discussions about the warming trends highlight the different warming trends and spatial patterns among the used GCMs.

p15 l2-5 The Watterson analysis is probably global, which may not be that meaningful for this Greenland application. This should be mentioned. MIROC5 often scores best when used with MAR compared to other GCMs. This could be mentioned, too.

Done. We mentioned that the Watterson analysis was performed on a global scale.

p15 l21 Remove "increased" before thickening, or was it thickening already?

We delete "increased".

p15 l23 Not sure comparison with observations really holds here. You mean marginal thinning and central thickening as large-scale features? Yes, but the thinning must reach much further inland here. Consider revising.

Done. We rewrite the sentence to: "Generally the large-scale pattern of marginal thinning and central thickening correlates with observations [...]"

p15 l25 Is there evidence from other studies that the 79 glacier is vulnerable? SMB changes in HadCM and MIROC seem very small here and the pattern of retreat looks almost identical. What happens in the unforced control run in this region? Are these changes in the figure also calculated relative to the control run (i.e. double differences)? They should! In any case it may be interesting to inspect and show the control run in a figure.

Done. We have deleted this sentence. The height changes in the Figures are not accounting the control run. We have updated them accordingly but they remain qualitatively the same.

p15 l33-34 Here the discussion of 2100 and 2300 changes is mixed with recent changes.

Done. We have deleted this sentence.

p16 Maybe "ice discharge anomaly"?

Done. As no line number is given we checked all occurrences on page 16 and changed where appropriate.

p17 What is "It"? Clarify.

For the IPSL-CM5A-LR experiment mentioned in the sentence before. We have rewritten this sentence.

Figure 1 The offset between GCMS in temperature in (b) does not correspond with the curves in Fig 3 because of the different reference periods. This makes comparison later in the manuscript difficult and may also have an impact on the timing of the overshoot.

See answer to MC3.

Figure 3 "according to Fig 4.". Replace "thick line" by "solid lines"

Done.

Figure 4 Remove or replace by a more meaningful comparison.
See answer to MC1.

Figure 5 Remove or replace by a more meaningful comparison.

See answer to MC1.

Figure 6 Are values trimmed at +-25? If yes, say so in the caption and give the min/max values.

Done. The values are trimmed.

Figure 8 Explain what the blue dots stand for in a.
Done.
Does an r^2 =1.00 mean perfect correlation? How is that possible?
The value was rounded from 0.99.. to 1.00.

Figure 9 Replace "Straight" by "Solid"

Done.

Figure 10 May want to add an estimate for the SMB-only contribution.

Done. We add a dashed line (0.4mm/a) calculated from RACMO.

[revised manuscript text omitted]

(a)                                          (b)

**Figure 9.** Present day velocities (year 2000) using SEMIC-HadGEM2: (a) observed velocities, (b) simulated velocities. Observed velocities: Rignot and Mouginot (2012).

[Figure]

**Figure 10.** Scatter plots of the present day state (year 2000) using the SMB forcing SEMIC-HadGEM2: (a) velocities, (b) ice surface elevation. Blue and red dots in (a) represent floating and grounded points, respectively. Observed velocities: Rignot and Mouginot (2012); Observed surface elevation: Morlighem et al. (2014). The gray line depicts the identity.

[Figure]

**Figure 11.** Sea level equivalent (SLE in mm) until the year 2100 (left panel) and 2300 (right panel)  under RCP2.6 forcing (Solid lines) and RCP2.6 forcing without overshoot (dotted-dashed). Additionally the control run (black dashed line) and the model mean and rms deviation from Fürst et al. (2015, Table B1) are shown. The  colored dots represent the onset years of overshooting 1.5°C in the global mean near-surface air temperature in a  30-year moving window relative to pre-industrial levels.

[Figure]

**Figure 12.** Lag (j) of projected sea level rise per year for three GCMs under RCP2.6 forcing (colored dots) and the modified RCP2.6 forcing without overshoot (colored circles) as mean for a time period similar to the observational period (2002–142002–2014). The black line indicates the observed value of 0.73 mm a$^{-1}$ by Rietbroek et al. (2016) .

[Figure]

**Figure 13.** Comparison of multi-year mean surface elevation ($h_s$) differences under RCP2.6 forcing between 2100-2000 (top row) and 2300-2000 (bottom row) for (a, d) SEMIC-HadGEM2, (b, e) SEMIC-IPSL and (c, f) SEMIC-MIROC5. The black contour line depicts the present-day ice mask. Positive values represent glacier thinning; negative values thickening. The data are clipped at ice thickness of 10 m (gray shaded area).

[Figure]

**Figure 14.** Comparison of multi-year mean surface velocity ($v$) differences under RCP2.6 forcing between 2100-2000 (top row) and 2300-2000 (bottom row) for (a, d) SEMIC-HadGEM2, (b, e) SEMIC-IPSL and (c, f) SEMIC-MIROC5. The black contour line depicts the present-day ice mask. Positive values represent glacier acceleration; negative values deceleration. The data are clipped at ice thickness of 10 m ( gray shaded area).

---

## Author Response (AR3)

Dear Editor,
we all want to thank you for the new reviews and the useful comments and suggestions! We want to emphasize, that the detailed points we received from Clemens were extremely useful and we want to express our gratitude for the work he has put into improving our manuscript. In the text below we provide a point to point answer and we carefully followed your instruction to go through the entire text to come up with lighter sentences and tried to do our best to make as easiest to understand as we can. Again, whenever our answer is 'done' we followed the suggestion to 100% in the new version of the text and in blue you will find our answers, while the review we received is shown in black color.
Throughout the entire review process, the reviews provided quite many new aspects of how modelers other than ice modelers may understand this or that procedure and allowed us to develop a new perspective on the different sections and subsections of the work. This has been incredibly useful also for discussions with scientists coming from atmospheric modelling and we want to thank all reviewers and the editor for pushing us into this direction!

Sincerely,
Martin and Co-authors

**Comments Editor**
"Due to the fact, " (p. 6 and p. 11). It is the first time I see a comma after fact. In both cases the sentence can be made lighter. E.g., p. 11: "As the inversion technique applied to for computing the friction coefficient ... ".
Done.

In the text: many instances, if not all, of "in order to" can be replaced by "to"
We delete this at various locations in the text.

The copy-editor will certainly provide further suggestions, but please read again your text while thinking about ways to write simpler or lighter sentences.
We went rigorously through the text, got further colleagues engaged in identifying too complex sentences and hope that the text is now much better. However, we are happy to receive any suggestions of the copy-editor later on.

Figure 2, legend "The gray line depicts the identity." -> "The gray line is the identity (or the identity function)"
Done.

Figure 4, legend: gray color : do you mean "gray shade" ?
Done. We replaced colour with shade.

**REVIEW Clemens Schannwell**
[...] I would like to urge the authors to go through the manuscript with a fine brush to iron out any language lapses. I am listing below all I could find, but it is quite likely that I missed a few.
see above – similar topic raised by the editor

**Scientific comments**
On page 5 in the paragraph lines 16-22, you are saying that SEMIC provides annual mean surface temperatures. Are you using these temperatures as a Dirichlet boundary condition at the surface for the temperature equation or are you using the near-surface temperatures from the GCMs? If the former, how do you account for intra-annual temperature variabilities? If the latter, do you correct the temperature for the height difference between the GCM and ISSM? Please clarify!
We use annual mean surface temperatures as Dirichlet BC at the surface. This is mentioned in the manuscript at P3L27, P9L13 and P5L22. To clarify this, we have rewritten the paragraph

around P9L13 slightly. Intra-annual variations of surface temperature are only covered in the SMB calculation and not in the temperature/enthalpy equation. To prescribe intra-annual variations within our ice model does not make sense, as we do not have a firn model that account for near surface processes. However, using the annual mean is a good approach as intra annual variations are smoothed out around 15m depth. Usually ISMs are driven by annual mean surface temperatures.

I am also still missing what kind of boundary condition you apply at the base of the floating parts for the temperature equation. I suspect that no matter what you specify there, it wouldn't affect your results much, but for completion it would be nice if you could add this piece of information to the second paragraph on page 10 (I think I would set a Dirichlet condition to the local pressure melting point temperature).
Yes, you are right the BC is missing in the text. We followed here the ISSM default settings given in Larour et al. (2012). We added: "At the base of floating ice we use a Neumann boundary condition that parameterizes the heat flux at the ice-ocean interface (Eq. 27 in Larour et al., 2012)."

Could you add somewhere in the inversion section, if you use any sort of regularisation (e.g. Tikhonov?) in your basal friction coefficient inversion? If so, how do you estimate the optimal parameters for the regularisation (usually done with an L-curve analysis but this might be a bit expensive for your advanced initialisation method)? If you do not use regularization, how to you ensure that you get a smooth velocity field?
We used a Tikhonov regularization but didn't perform an L-curve analysis. For the parameters, we followed the inversion approach given by Seroussi et al. (2013) for modeling the GrIS. We added to the text: "The cost function is composed of two terms which are fitting the velocities in fast- and slow-moving areas. A third term is a Tikhonov regularization to avoid oscillations. The parameters for weighting the three contributions to the cost functions are taken from Seroussi et al. (2013). "

**Specific comments**
I am noting all typos and grammar issues I found but the authors need to go through the manuscript with a fine brush and give it a good workover in terms of language style in general. Also I am not sure what the ESD policy is on whether North American or British spelling should be used, but in this manuscript it is mostly a strange mixture e.g. coloured (BE), but initialized (AE).
We followed here the Oxford spelling for BE, where the suffix –ize can be used instead of –ise (https://en.wikipedia.org/wiki/Oxford_spelling).

Throughout the manuscript it is **sea level**, but **sea-level rise**.
Done. Changed sea level rise to sea-level rise.

P1L5 approach between = approach of
Done.

P1L7 Just to check, are citations allowed in the abstract? I don't find it necessary here as you cite Krapp et al. 2017 later in the text.
We have not found anything about the in the ESD guidelines. But we agree, it is not necessary in the abstract and dropped it there.

P1L13 Here and throughout spatial-integrated=spatially-integrated
Done

P1L12-14 please replace "never falls below zero" with something along the lines of "SMB remains positive". Also "a recovery of SMB towards values of slightly below present day" is a bit to general for my taste. You have to know that the SMB is positive for Greenland to understand this. This is OK for the main paper where you give numbers, but if the SMB for the whole ice sheet was

negative, this sentence would sound strange. Why instead not write that SMB "decreases in the latter half of the simulation period reaching values similar to present day"?
Done. We have rewritten the sentence to: "This is due to a spatially-integrated SMB that remains positive and reaches in the latter half of the simulation period values similar to present day."

P2L1 This sentence makes no sense. Are you trying to say that Greeland's contribution to global sea-level rise has been 20%? Please rephrase.
Yes, exactly that's what we want to say. We have deleted the typo in the sentence.

P2L2 delete "to global sea-level rise"
Done.

P2L4 Nerem et al. 2018 show only that SLR is accelerating, but not why. Please use a different reference.
Done. We use: Enderlin et al. (2014).

P2L6 keep a global = keep the global
Done.

P2L19 by that time = by 2100
Done.

P2L19 by 2000 and ... should that not read 2100?
Yes, you are right. Changed 2000 to 2100.

P2L26 delete "abated"
Done.

P3L6 add at the end: ", which is employed/used in this study."
Done.

P3L6-7 delete "by means"
Done.

P3L16 make sure you introduce the acronyms at first use. You don't seem to introduce them in the abstract. Please make sure this conforms with the ESD guidelines.
The ESD guideline says: They need to be defined in the abstract and then again at the first instance in the rest of the text. So, ISSM is now introduced in the abstract. Beside HadGEM2-ES, IPSL-CM5A-LR and MIROC5 all abbreviations are introduced in the abstract and first appearance in the text.

P3L30 delete "very"
Done.

P4L1 based on reconstruction. What reconstruction?
We have rewritten the sentence to: " … performed an optimization for the GrIS forced with regional climate model data (MAR)."

P4L10 parentheses around SMB missing
Done.

P4L16 from available energy = from the available energy
Done.

P4L31-P5L1 MIROC5 also exceeds this threshold according to Figure 1
Done. MIROC5 is also mentioned.

P5L3 it stabilizes= temperatures stabilize
Done.

P5L9 delete second "the"
Done.

P6L8 parameter=parameters
Done.

P6L13 delete "However"
Done.

P6L18 still lacks details and quality compared = still lack details compared
Done.

P6L21 differs up to = differs by up to
Done.

P6L21 full stop after 200 Gt a$^{-1}$
Done.

P7L20-21 Delete sentence starting with "This one way coupling ...". You have said this before.
Done.

P7L28 delete focusing on negative emissions
Done.

P7L30-31 Delete sentence starting with "As mentioned before ...". You have said this before.
Done.

P8L13 delete "over the year from SMB"
Done.

P8L22 below the magnitude of present day = present day magnitude
Done.

P8L27 among the models is rather similar = is rather similar among the models
Done.

P8L29 agrees well to = agrees well with
Done.

P8L31 delete sentence. Try to avoid hollow statements like this.
Done.

P8L32 change sentence structure to read: "For the available RACMO2.3 time series and the SEMIC-GCMs we have computed the interannual SMB variability.
Done.

P9L7 year missing from citation
Done.

P11L13 please write out approximately here and throughout
Done.

P11L18 remove parentheses around Rignot and Mouginot citation

Done.

P11L20-24 Out of curiosity, can you restart ISSM simulations from different 3D unstructured grids? Is there a routine that interpolates these fields between the meshes?
Yes, there are routines to interpolate 2d and 3d unstructured meshes that come along with ISSM. For this study, we kept the number of vertical layers constant for each grid sequence. We then employ the 2d routine for each layer separately. For 3D we have developed our own routine.

P13L16 I am not a native speaker, but I don't think I would capitalise compass directions unless I talk about a specific region e.g. Western Rockies. But I leave this up to the editor.
Ok, we found the same in the ESD guidelines. We changed the capitalization where necessary.

P14L11 less=lower
Done.

P14L23 experienced = experiences
Done.

P14L24 change to something like "mass loss from a decline/less positive surface mass balance"
Done.

P16L12 change to "response is different across all models"
Done.

P16L16 delete "abated"
Done.

P17L12 whether=neither
Done.

P18L12 change to "Despite all three GCMs being based on ...."
Done.

P18L13 instead of considerably large, please give a number and avoid such vague language.
Instead of 'considerably large' we use 'different' here. Numbers are given in the next sentence.

Figure 4 Is there a solid black line in the upper panel? Might be worth it to bring the black line to the front in both panels.
We agree that the black line is hard to see. But the black line and dark grey shaded box are belonging to each other (mean value and variability from polarportal, respectively), that's why we kept it in the same color range. Plotting these data on top will overlay the GCM data, which we do not want. If the typesetting/production will give us some instructions to improve it, if the quality is not acceptable, we are happy to revise the colors again.

Figure 5 alls = all.
Done.

The thick solid lines show 30-year moving means. (all lines are solid in this figure!)
Done. We have rewritten this to: "The thick lines are a 30-year moving mean calculated from the yearly data (thin lines)."

Figures 7 & 8 Why are the fields from IPSL so patchy? Is this a resolution issue?
This is indeed very strange. For HadGEM and MIROC the patchy pattern disappears after the lapse rate corrections. In IPSL, however, the patchy pattern remains. The only explanation for this is the native grid resolution, which is quite coarse for IPSL in the latitude direction:

IPSL: 2.5° x 1.5° (lat x lon)
HadGEM: 1.25° x 1.875° (lat x lon)
MIROC5: 1.4° x 1.4° (lat x lon)

Figure 9 In caption: how can this be observed velocities if HadGEM2-ES was used? Can we also have different colourbar labels? They look odd.
Observed velocities are assembled from satellite radar interferometry. It is an independent product from the SMB forcing provided from Rignot & Mouginot (2012). Figure 9b shows this data product. Figure 9a displays the ice surface velocities from ISM forced with SEMIC-HadGEM2-ES at the year 2000.
The figure now is redesigned, i.e. new limits of colorbar, new labels at colorbar and colorbar is moved to the righthandside.

Figure 12 Caption change to: The solid black line indicates ...
Done.

**The effect of overshooting 1.5°C global warming on the mass loss of the Greenland Ice Sheet**

Martin Rückamp[1], Ulrike Falk[2], Katja Frieler[3], Stefan Lange[3], and Angelika Humbert[1,4]

[1]Alfred Wegener Institute, Helmholtz Centre for Polar and Marine Research, Bremerhaven, Germany
[2]formerly Alfred Wegener Institute, Helmholtz Centre for Polar and Marine Research, Bremerhaven, Germany
[3]Potsdam Institute for Climate Impact Research, Potsdam, Germany
[4]University of Bremen, Bremen, Germany

*Correspondence to:* martin.rueckamp@awi.de

**Abstract.**  Sea-level rise associated with changing climate is expected to pose a major challenge for societies. Based on the efforts of COP21 to limit global warming to 2.0°C or even 1.5°C by the end of the 21[th] century (Paris Agreement), we simulate the future contribution of the Greenland ice sheet (GrIS) to  sea-level change under the low emission representative concentration pathway (RCP) 2.6 scenario. The ice sheet  system model (ISSM) with higher order approximation is used and initialized with a hybrid approach  of spin-up and data assimilation. For three general circulation models (GCMs: HadGEM2-ES, IPSL-CM5A-LR, MIROC5) the projections are conducted up to 2300 with forcing fields for surface mass balance (SMB) and ice surface temperature ($T_s$) computed by the surface energy balance model  of intermediate complexity (SEMIC). The projected  sea-level rise ranges between 21–38 mm by 2100 and 36–85 mm by 2300. According to the three GCMs used,  global warming will exceed 1.5°C  early in the 21[th] century. The RCP2.6 peak and decline scenario is therefore  manually adjusted in another set of experiments  to suppress the 1.5°C-overshooting effect. These scenarios show a  sea-level contribution that is on average about 38% and 31% less by 2100 and 2300, respectively.  For some experiments, the rate of mass loss in the 23[rd] century  does not exclude a stable ice sheet in the future. This is due to a  spatially-integrated SMB that remains positive and reaches values similar to the present day in the latter half of the simulation period. Although the mean SMB is reduced in the warmer climate, a future steady-state ice sheet with lower surface elevation and hence volume might be possible. Our results indicate  that uncertainties in the projections stem from the underlying GCM climate data used to calculate the surface mass balance. However, the RCP2.6 scenario will lead to significant changes of the GrIS, including elevation changes of up to 100 m. The  sea-level contribution estimated in this study may serve as a lower bound for the RCP2.6 scenario, as the  currently observed sea-level rise is not reached in any of the experiments; this is attributed to processes (e.g. ocean forcing) not yet represented by the model, but proven to play a major role in GrIS mass loss.

*Copyright statement.* We agree to the copyright statements given on the webpage of ESD. The figures within the manuscript are produced by the authors and have not been published by the authors or others in other journals.

[revised manuscript text omitted]

where $\text{SMB}_{\text{dyn}}(x,y,t)$ and $\text{SMB}_{\text{fix}}(x,y,t)$ are the SMB values with and without taking height changes into account, respectively. The surface elevation changes are taken from the ISSM elevation $h_s(x,y,t)$ while running the simulation and a reference elevation $h_{\text{fix}}(x,y)$. In our setup the reference elevation  corresponds to the ISSM ice surface elevation at the initial state.

In this parameterization the SMB gradient $b_i$ is dependent  on both location and sign. It can take four values and a separation is made on the location relative to 77°N and on the sign of the SMB. This separates regions of largely different sensitivity, namely the ablation zone with a larger gradient compared to the accumulation zone, and a more sensitive ablation zone in the  south compared to the  north. While a complete uncertainty analysis is given by Edwards et al. (2014a), only the maximum likelihood gradient set, $\boldsymbol{b} = (b_p^N, b_n^N, b_p^S, b_n^S)$, is used here:

$$b_p^N = 0.085\,\text{kg}\,\text{m}^{-3}\,\text{a}^{-1},$$
$$b_n^N = 0.543\,\text{kg}\,\text{m}^{-3}\,\text{a}^{-1},$$
$$b_p^S = 0.063\,\text{kg}\,\text{m}^{-3}\,\text{a}^{-1},$$

[revised manuscript text omitted]